# Selective translation of epigenetic modifiers affects the temporal pattern and differentiation of neural stem cells

Quan Wu [1✉], Yuichi Shichino [2], Takaya Abe [3], Taeko Suetsugu[1], Ayaka Omori[1], Hiroshi Kiyonari [3], Shintaro Iwasaki [2,4] & Fumio Matsuzaki [1,5✉]

The cerebral cortex is formed by diverse neurons generated sequentially from neural stem cells (NSCs). A clock mechanism has been suggested to underlie the temporal progression of NSCs, which is mainly defined by the transcriptome and the epigenetic state. However, what drives such a developmental clock remains elusive. We show that translational control of histone H3 trimethylation in Lys27 (H3K27me3) modifiers is part of this clock. We find that depletion of *Fbl*, an rRNA methyltransferase, reduces translation of both Ezh2 methyltransferase and Kdm6b demethylase of H3K27me3 and delays the progression of the NSC state. These defects are partially phenocopied by simultaneous inhibition of H3K27me3 methyltransferase and demethylase, indicating the role of Fbl in the genome-wide H3K27me3 pattern. Therefore, we propose that Fbl drives the intrinsic clock through the translational enhancement of the H3K27me3 modifiers that predominantly define the NSC state.

---

[1] Laboratory for Cell Asymmetry, RIKEN Centre for Biosystems Dynamics Research, Kobe, Hyogo 650-0047, Japan. [2] RNA Systems Biochemistry Laboratory, RIKEN Cluster for Pioneering Research, Wako, Saitama 351-0198, Japan. [3] Laboratories for Animal Resource Development and Genetic Engineering (LARGE), RIKEN Centre for Biosystems Dynamics Research, Kobe, Hyogo 650-0047, Japan. [4] Department of Computational Biology and Medical Sciences, Graduate School of Frontier Sciences, The University of Tokyo, Chiba 277-8561, Japan. [5] Laboratory of Molecular Cell Biology and Development, Department of Animal Development and Physiology, Graduate School of Biostudies, Kyoto University, Kyoto, Japan. ✉email: quan.wu@riken.jp; fumio.matsuzaki@riken.jp

How the developmental schedule is shared by all individuals in a given species of animals is a fundamental question in developmental biology. One fascinating hypothesis is the presence of a developmental clock that counts time for the developmental program, and several potential mechanisms have been proposed that could work in this manner. During somitogenesis, a clock consisting of a complex gene regulatory network generates oscillations and regulates segmentation in a defined time[1]. The cell cycle is also an oscillator that counts the time to initiate transcription of the zygotic genome during the midblastula transition of *Xenopus* and *Drosophila*[2,3]. In oligodendrocyte precursors of rat optic nerve, an hourglass type clock was observed: an accumulated amount of a CDK inhibitor p27/kip1 during proliferation of oligodendrocyte precursors determines the timing for their differentiation[4]. Moreover, an epigenetic clock based on DNA methylation is a promising predictor of biological age[5].

The sequential generation of diverse neurons from a small population of neural stem cells (NSCs) in a highly orchestrated order in the mammalian cerebral cortex may also be controlled by a developmental clock. Following proliferation, NSCs divide asymmetrically to produce one stem cell and either a neuron or intermediate progenitor, the majority of which divide once before terminal differentiation[6,7]. As neurogenesis proceeds, a shift in NSC gene expression, or identity, occurs; thus, NSC identity is temporally patterned and initiates the production of a diverse array of neuronal progeny. NSCs initially produce deep-layer neurons (early-born), followed by upper-layer neurons (late-born), and finally glia[8,9].

Temporal patterning of NSC identity is widely observed among species, from the central brain of *Drosophila*, to the mammalian retina, implying the existence of a conserved strategy for neuronal production[8,10]. In *Drosophila*, key temporal determinant genes have been identified, and epigenetic mechanisms are involved in regulating these genes. For example, NSC expression of the *Hunchback* gene is epigenetically restricted by the relocation of the *Hunchback* locus into a repressive subnuclear compartment[11]. In the mammalian cortex, a set of temporal genes concordant with the progression of temporal identity has been identified.[10,12]. Furthermore, epigenetic modifiers have been shown to play a key role in temporal patterning in mammalian NSCs. For example, polycomb repressive complex 2 (PRC2), consisting of a histone methyltransferase enzyme Ezh2, Suz12, Eed, and RbAp48, is an epigenetic modifier of histone H3 trimethylation at Lys27 (H3K27me3). Perturbation of PRC2 function compromises temporal shifts in NSC identity, leading to disordered production of NSC progeny cells[12–15]. In the embryonic forebrain, a demethylase of H3K27me3, Kdm6b, is required for NSCs to activate neurogenic genes in response to retinoic acid[16]. In embryonic stem cells, Kdm6b is essential for neural commitment[17], whereas its function in temporal fate changes of NSCs has not been clarified. Thus, the precise temporal pattern of NSC gene expression significantly depends on the precise control of temporal genome-wide epigenetic modifications. Thus, how genome-wide epigenetic modification goes forward over time needs to be elucidated. Then, critical questions are whether it constitutes a part of the developmental clock acting on NSCs, and if this is the case, what mechanisms operate to drive this clock. Here, we show that Fbl, an rRNA methyltransferase, is required for temporal progression and differentiation of NSCs. Fbl potentially drives the developmental clock by promoting the translation of the key modifiers of H3K27me3 in NSCs.

## Results

### Analysis of brain development at the single-cell level.
We first investigated temporal identity changes of NSCs by performing transcriptome analysis at the single-cell level from embryonic (E) day 11, in which mostly of NSCs are proliferative or at the early neurogenic stage, to E14, the period producing late-born neurons at the mid-neurogenic stage. We performed a single-cell transcriptome analysis for different genotypes along the developmental timeline using the 10X genomics single-cell RNA-seq kit (Fig. 1a; see "Methods" for sample collection). Briefly, each cell was mixed with a functional gel bead containing reverse transcription reagents in a droplet. In each droplet, the cell was lysed and mRNAs from the cell were reverse-transcribed by priming with a ploy-T primer, which also contains a specific cell barcode for cell identification and a unique molecular index (UMI) for mRNA identification. After PCR amplification and size selection, the constructed library was sent for sequencing. We first compared our wild-type data at different stages (E11, E12, E13, and E14) with previously published data at E11, E13, and E15[18]. We detected more genes from each cell (2000–4000 genes/cell) and more copies of mRNA (0–20,000 copies/cell) compared with their data (1000–2000 genes/cell; 0–10,000 copies/cell). The percentages of reads that mapped to the mitochondrial genome, which might be related to low-quality or dying cells, were similar and acceptable (<10%) in both datasets (Supplementary Fig. 1a, b). We also investigated the expression of the pattern of several marker genes in both datasets and found that both datasets showed similar expression patterns for all these genes (Supplementary Fig. 1c–h). These results indicated that our transcriptome data are of high quality enough to perform temporal analyses at the single-cell transcriptome level.

We then constructed a continuous trajectory of all cells including NSCs, neural progenitors, and neurons by a force-directed k-nearest neighbor graph using SPRING[19], by which cells with similar gene expression patterns are clustered together and cells which are transiting from one cluster to another are plotted between these clusters. We then interpreted the cell clusters on the basis of known markers and stages. Consistent with a previous model, the expression of early- (Hmga2+) and late-onset genes (Zbtb20+) in NSCs was correlated with the timing for production of deep- (Tbr1+) and late-neuron (Satb2+), respectively (Fig. 1b, c). Thus, NSCs gradually change their transcriptome to produce different types of neurons[10].

### Identification of Fbl as a candidate regulator of temporal patterning.
We then searched for factors that were associated with the progression of the developmental clock and the transition of the temporal identity of NSCs. We assumed the presence of genes showing monotonic changes of expression among the factors promoted the temporal pattern as a simple hypothesis. Therefore, we compared our single-cell transcriptome data from E11 and E14 NSCs[10]. To categorize gene transcriptome profiles according to the pattern of their expression changes from E11 to E14, we used weighted correlation network analysis (WGCNA)[20], an analytical method by which genes are classified into "modules" (sets of genes); genes in the same module show a similar temporal pattern of expression to each other. We identified a gene module with higher expression in E11 than in E14 NSCs (brown module in Supplementary Fig. 2a) that was highly enriched in genes whose products are located in nucleoli and their fibrillar center; nuclear regions essential for rDNA transcription and pre-rRNA processing and ribosomal protein synthesis. (Fig. 1d, e and Supplementary Data 1). These results raise the possibility that translational regulation is important for temporal fate transitions of NSCs. To test this hypothesis, we particularly focus on Fbl (also known as Fibrillarin) as an example to investigate the impact of translational regulation during this process. Fbl was initially reported as an rRNA methyltransferase for 2′-O-methylation and plays an essential role in the development and disease[21–24].

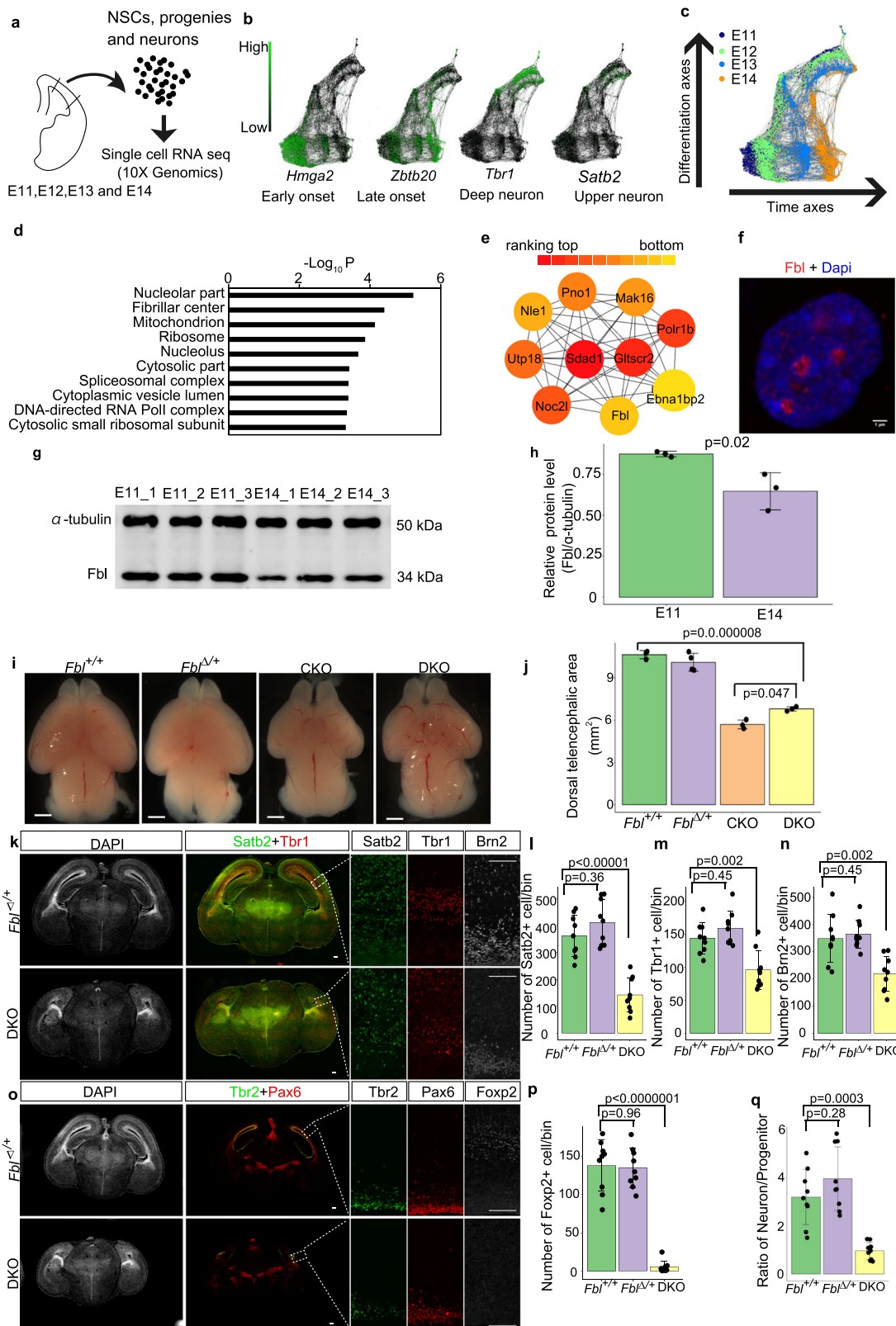

Although Fbl is regarded essential for the translational regulation of some mRNAs[25], its roles and underlying mechanisms in mouse brain development are unclear. To address these issues, we first investigated the expression pattern of Fbl. The nucleolar expression of Fbl in all NSCs and neurons in E11 and E14 was observed by immunostaining (Fig. 1f and Supplementary Fig. 2b). Using western blotting, we found higher Fbl protein levels in E11 than

E14 FACS-isolated NSCs from *Hes1-d2-EGFP* reporter mice, in which d2-GFP is expressed under the control of the promoter of an established NSC marker: *Hes1*[26] (Fig. 1g, h).

**Knockout of *Fbl* disrupted brain development independently of apoptosis**. To examine the function of Fbl in NSCs, we

**Fig. 1 Fbl is essential for brain development. a** Experiment design for single-cell (sc) RNA-seq. Cell number in scRNA were E11 $n = 829$, E12 $n = 2457$, E13 $n = 1566$, and E14 $n = 1293$. **b**, **c** SPRING graph of single cells colored by different stages (**c**), and expression patterns of an early-, a late-onset gene, a progenitor, and a neuron marker, respectively. **d** Gene Ontology (GO) analysis of the gene module (brown module) that showed higher expression in E11 than E14 neural stem cells (NSCs). **e** Top ten module nodes based on protein interaction networks in the brown module. **f** E14 NSC stained for Fbl and Dapi showing nucleolar Fbl expression ($n = 5$). Scale bars, 1 μm. **g** Western blot of Fbl and α-tubulin using isolated Hes1+ NSCs. **h** Quantitative analysis of Fbl protein levels showing reduced Fbl at E14 (two-sided Student $t$ test; data are presented as mean ± s.d. of $n = 3$ experiments). **i** Whole-mount brain image at E17 showing microcephaly after Fbl knockout. The genotype of mice are: Fbl + /+:Fblflox/+P53−/−;FblΔ/+:Fblflox/+ P53−/−Emx1Cre/+; Fblflox/ +P53 + /−Emx1Cre/+; CKO:Fblflox/floxP53 + /−Emx1Cre/+; DKO:Fblflox/floxP53−/− Emx1Cre/+. Scale bars, 1 mm. **j** Quantitative analysis of the dorsal telencephalic area in different mice. Data are presented as the mean ± s.d. of $n = 3$ or 4 brains Statistic analysis was performed by one-way ANOVA followed by Tukey's post hoc tests. **k**, **o** E17 brain sections showing a reduced number of both deep- and upper-neurons in DKO. Scale bars, 100 μm. **l–n**, **p** Immunostaining-based cell number quantification ($n = 3$ mice per genotype, $n = 3$ sections per mouse). **q** Ratio of Tbr1+ or Brn2+ neurons and Pax6+ or Tbr2+ progenitors ($n = 3$ mice per genotype, $n = 3$ sections per mouse). Data are presented as the mean ± s.d. of $n = 9$ sections. Statistic analysis was performed by one-way ANOVA followed by Tukey's post hoc tests. For each section, an area with a length of 200 μm along with an apical membrane was counted. Such region was defined as a bin.

conditionally deleted *Fbl* (the entire gene) in the developing dorsal cortex of E9.5 by crossing with *Emx1Cre* mice, in which the *Emx1* gene was replaced by *Cre* gene[27] (*Fbl* CKO; Supplementary Fig. 2c). We used heterozygous of *Emx1Cre* mice, because homozygous mutation of *Emx1* disrupts the corpus callosum[28]. The CKO mice showed microcephaly and dramatic brain size reduction, and died around postnatal day (P) 40 (Fig. 1i, j and Supplementary Fig. 2d, e). Microcephaly in *Fbl* CKO could be induced by caspase-mediated apoptosis of NSCs, as earlier work reported upregulated expression of *Trp53* and subsequent apoptosis in *Fbl* knockdown mouse embryonic stem cells[29]. Indeed, we detected high levels of cleaved caspase-3 (CASP3) expression in E12.5 *Fbl* CKO brains (Supplementary Fig. 2f). To determine whether Trp53-dependent apoptosis is responsible for microcephaly, we crossed *Fbl* CKO and *Trp53*−/− mice. As *Trp53* knockout did not affect brain size, nor neuron number compared to wild type (Supplementary Fig. 3a–c), we used wild type or heterozygous mice for *Fbl* as controls (designated as $Fbl^{+/+}$ or $Fbl^{\Delta/+}$ in Fig. 1i; a subset of these mice, which displayed exencephaly and anencephaly because of the failure of neural tube closure, were excluded from our analysis[30]). We obtained double-knockout mice with genotype *Fblflox/flox, Trp53*−/−, *Emx1Cre/*+ (DKO) in which we confirmed the loss of Fbl by immunohistochemistry (Supplementary Fig. 2g). We also detected several rRNA sites with reduced methylations upon deletion of *Fbl*, consistent with a previous study showing that Fbl is a methyltransferase of rRNA[25] (Supplementary Fig. 3d; see "Methods" for a description of the method). DKO brains were smaller than control brains, although apoptosis was completely suppressed, indicating that microcephaly could not be explained by NSC apoptosis alone (Fig. 1i, j).

Next, we tested the possibility that premature differentiation of NSC and defective neurogenesis could cause microcephaly in *Fbl*-lacking mice by investigating DKO cortical organization at a late neurogenic stage (E17). Notably, we observed a significant decrease in the number of both deep-layer (Tbr1+ or Foxp2+) and upper-layer neurons (Satb2+) in DKO brains compared with $Fbl^{+/+}$ or $Fbl^{\Delta/+}$ mice using immunohistochemistry (Fig. 1k–m, o, p). Moreover, the cell population expressing *Brn2*, a crucial gene for the production of upper-layer neurons[31], was also reduced in DKO brains (Fig. 1k, n). Furthermore, we observed a significant reduction in the number of Olig2+ oligodendrocytes and of cells expressing Zbtb20, which is essential for astrogenesis[32] in DKO mice (Supplementary Fig. 3e–g). If premature differentiation caused a decrease in neurons, the ratio of neurons to progenitors should be increased at late neurogenesis. However, this was not the case (Fig. 1o, q), indicating that the *Fbl*-deleted mouse cortex had defective neurogenesis that was not caused by premature NSC differentiation.

To directly evaluate the role of Fbl in NSCs, we knocked down *Fbl* using siRNA. The siRNA treatment reduced *Fbl* mRNA to around 30%. In this situation, we did not observe the upregulation of the apoptosis-related gene: *Trp53* (Supplementary Fig. 4a). We then performed RNA-seq analysis and found that the persistent of deep-layer makers (*Bcl11b* and *Tbr1*) and delay of expression of upper-layer markers (*Pou3f2, Pou3f3, Satb2,* and *Zbtb20*), suggesting the delay of fate transition after knockdown of *Fbl* (Supplementary Fig. 4b, c and Supplementary Data 2a).

**Analysis of temporal identity of *Fbl*-mutant NSCs at the single-cell level**. To clarify the possible mechanisms leading to defective neurogenesis in DKO, we performed a single-cell transcriptome analysis for different genotypes along the developmental timeline using the 10X genomics single-cell RNA-sequencing kit. We analyzed these transcriptome data using t-distributed stochastic neighbor embedding (t-SNE) clustered the cells according to developmental time and cell type (Supplementary Data 2b showing specific markers for each cluster). At E10 or E12, the DKO cells were merged with $Fbl^{+/+}$ or $Fbl^{\Delta/+}$ cells (the central clusters and lower-left clusters in Supplementary Fig. 5a), however, at E14, the DKO cells were clearly separated from $Fbl^{+/+}$ or $Fbl^{\Delta/+}$ cells (cell clusters surrounded by a black-dotted line and clusters indicated by a red dotted line in Supplementary Fig. 5a). We detected that differentially expressed genes (DEGs, fold change > 0.25, $q$-value < 0.01) between $Fbl^{\Delta/+}$ and DKO NSCs (cells in cluster 0, 1, 5, 7, 11, 12 in Supplementary Fig. 5b) and found that the total number of DEGs was increased from 41 (E10) to 74 (E12), and 760 (E14) (Supplementary Data 2c–e shows the DEGs list). These observations indicate that dramatic transcriptome changes occurred in DKO NSCs compared to $Fbl^{\Delta/+}$ control NSCs after E12.

Next, we investigated the properties of DEGs between $Fbl^{\Delta/+}$ and DKO NSCs at E14. NSCs gradually change their identity to produce different types of neurons (Fig. 2a). Consistent with this model, PCA organizes the cells from E11 and E14 dorsal brains into two directions: a differentiation axis (PC1) and a temporal axis (PC2) (Fig. 2b). We computed the contributions of each of the DEGs to PC1 and PC2, which represent the relevance of the gene to each axis, and compared them with those of randomly selected genes as a control set (Supplementary Data 2f). We found that the contributions of DEGs to PC1 and PC2 were significantly different from those of randomly selected genes, suggesting that deletion of *Fbl* affects both differentiation and temporal axis (Fig. 2c).

We then asked whether Fbl promotes or suppresses the progression of NSCs along these two axes. To answer this question, we again constructed a continuous trajectory of all cells

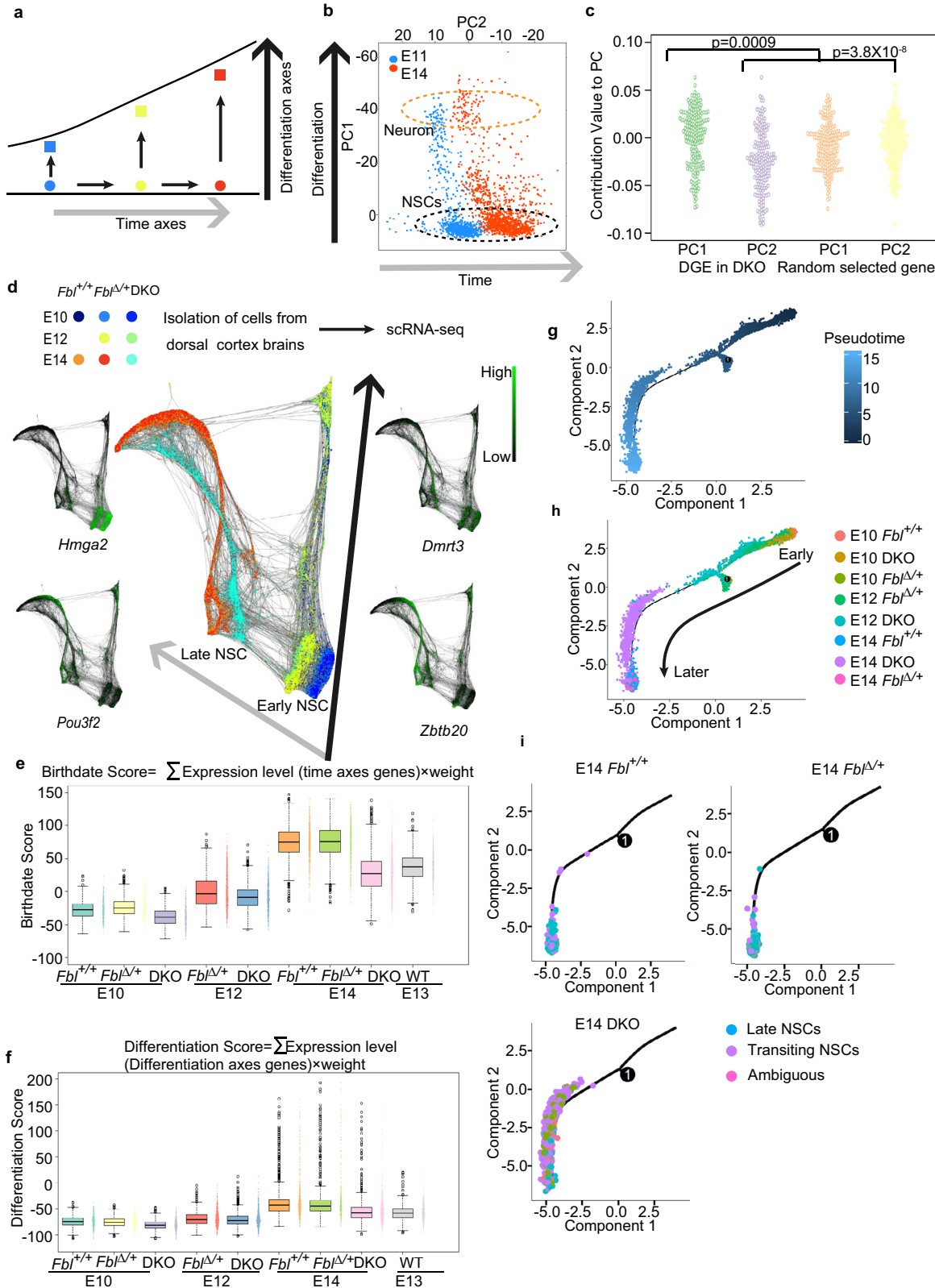

with SPRING[19] (Supplementary Fig. 5c). Therefore, cells were deposited along the differentiation and temporal axes. Consistent with the t-SNE analysis, DKO cells from E10 and E12 could not be distinguished from controls at the same stages. In contrast, E14 DKO cells were plotted between E12 *Fbl*$^{\Delta/+}$ and E14 *Fbl*$^{+/+}$ or *Fbl*$^{\Delta/+}$ cells, implying delayed temporal identity transition (Fig. 2d).

To further confirm our results, we introduced a simple mathematical model to estimate the developmental time and differentiation state of each NSC (see "Methods" for detail). We defined the birthdate score and the differentiation score of each cell as a weighted linear combination of specific temporal-axis and differentiation-axis genes, respectively[12]. For example, the birthdate score of a cell is a sum of the weight of a temporal

**Fig. 2 Single-cell transcriptome analysis of temporal patterning in neural stem cells (NSCs). a** Model of NSC temporal identity progression. **b** Principal components analysis (PCA) of transcriptome from E11 ($n = 846$) and E14 cells ($n = 1293$) organizes the cells on two axes: differentiation axis (PC1) and time axis (PC2). Orange circle: neurons; black circle: NSCs. **c** PC contribution of differentially expressed genes (DEG) between E14 DKO and Fbl$\Delta/+$ NSCs showing that Fbl affects both the differentiation axis and the time axis (Kruskal–Wallis test and Dunn's test with Bonferroni correction). **d** SPRING graph of single cells colored by genotype from different stages, and expression patterns of several early- and late-onset genes. The gray and black arrows represent temporal axes and differentiation axes, respectively. Different colored dots represent different stages or genotypes of cells. E10 Fbl$+/+$ $n = 836$, E10 Fbl$\Delta/+$ $n = 1202$, E10 DKO $n = 651$, E12 Fbl$\Delta/+$ $n = 2070$, E12 DKO 2260, E14 Fbl$+/+$ $n = 3905$, E14 DKO $n = 2879$, E14 Fbl$\Delta/+$ $n = 2592$. **e, f** Scoring single-cell identity with a mathematical model. Only NSCs that identified according to t-distributed stochastic neighbor embedding (t-SNE) analysis were used in (**e–h**) (see Supplementary Fig. 5b). E10 Fbl$+/+$ $n = 691$, E10 Fbl$\Delta/+$ $n = 982$, E10 DKO $n = 581$, E12 Fbl$\Delta/+$ $n = 1228$, E12 DKO $n = 1183$, E14 Fbl$+/+$ $n = 1046$, E14 DKO $n = 1480$, E14 Fbl$\Delta/+$ $n = 577$. Data are presented as box–whiskers (left) and bee swarm plots (right). Data in the box–whiskers show maximum (except outliers), third quartile to the first quartile, median, and minimum (except outliers). Outliers data are plotted as points outside of the box. Outliers are defined as 1.5 folds larger or smaller than interquartile range from third quartile or first quartile, respectively. Pseudotime alignment of NSCs via Monocle. The high-dimensional transcriptome data of each cell was reduced into two components by t-SNE. **g–i** Pseudotime alignment of NSCs via Monocle. The high-dimensional transcriptome data of each cell was reduced into two components by t-SNE. The dark-blue and light-blue dots in (**g**) represent cells in early and late pseudotime stages. Different colored dots represent different stages or genotypes of cells in (**h**). The positions of E14 cells with different genotypes are emphasized in (**i**). The blue, purple, and pink dots represent NSCs in late, transiting to late and unknown stages, respectively.

gene multiplied by its normalized expression level for all temporal genes (96 genes). The weight of a temporal gene is decided by the stage-dependent expression level and specificity of such gene; a gene that is higher expressed in late NSCs than early NSCs in most of late NSCs has a higher weight. These scores are likely a faithful representation of each cell, as both birthdate and differentiation scores increased from E10 to E14. The birthdate scores of E14 DKO NSCs were lower than those of the E14 $Fbl^{+/+}$ or $Fbl^{\Delta/+}$ NSCs, but similar to those of the E13 $Fbl^{+/+}$ NSCs (Fig. 2e). In addition, E14 DKO NSCs were less differentiated than E14 $Fbl^{+/+}$ or $Fbl^{\Delta/+}$ (Fig. 2f). These results suggest that Fbl has dual functions required for temporal progression and differentiation of NSCs. We also confirmed that knockdown of $Fbl$ led to a lower birthdate score (Supplementary Fig. 4d).

We also performed pseudotime analysis of NSCs; when NSCs are ordered such that any two cells are located next to each other if their transcriptomes are closer. This pseudotemporal ordering of NSCs from these stages suggested a delay in the temporal patterning of E14 DKO NSCs (Fig. 2g–i). In fact, immunohistochemistry confirmed the persistence of an early-onset gene, Dmrt3 in the E14 $Fbl^{\Delta/\Delta}$ dorsolateral cortex, while it almost disappeared in E14 $Fbl^{+/+}$ or $Fbl^{\Delta/+}$, and production of late-born neurons was also lower in the E14 DKO brains than in E14 $Fbl^{+/+}$ or $Fbl^{\Delta/+}$ (Supplementary Fig. 5d–f). All these results are consistent with the notion that the absence of Fbl results in a delayed progression of the temporal identity of the NSCs. Thus, we conclude that Fbl is required for the proper temporal patterning of NSCs.

**Fbl affected cell-cycle progression.** Next, we examined whether Fbl affected cell-cycle progression by measuring the incorporation of 5-ethynyl-2'-deoxyuridine (EdU) into NSCs. EdU pulse-labeling of S-phase cells for 1 h and immunostaining of M-phase cells with anti-phospho-histone 3 (pH3) antibody revealed significant reductions of both S-phase and M-phase cell populations in DKO at the E14 compared to control (Fig. 3a–c). To further investigate this cell-cycle defect, we analyzed the DNA content of NSCs using FACS after siRNA-dependent $Fbl$ knockdown (Fig. 3d, e). A significant increase in NSCs was observed in the G1/G0 phase and a reduction in S-phase NSCs 2 days after $Fbl$ knockdown. This suggests that Fbl affects the initiation of the S-phase (Fig. 3f), while this cell-cycle defect is unlikely to be a cause of a delay in the temporal pattern in the absence of Fbl, as shown below.

**Fbl on temporal identity transition is cell-autonomous but not via cell cycle progression.** The transition in NSC identity that promotes the shift to late-born neurons from early-born neurons requires feedback interaction between NSCs and early-born neurons[33]. However, if NSCs can receive feedback from neighboring early-born neurons, the temporal pattern of these NSCs proceeds normally, even though their own cell cycle is artificially arrested[10]. In the case of DKO, defects in temporal patterns might come from compromised feedback from early-born neurons. We then examined how $Fbl$-deleted cells are affected in the presence of early-born neurons in two different conditions: sparse culture of $Fbl$-deleted cells with surrounding normal cells and in vivo $Fbl$ deletion in a sparse population using CRISPR/Cas9. In both cases, $Fbl$-deleted cells produced less late-born neurons than control cells (Supplementary Fig. 6). These results indicated that defective Fbl cell-autonomously compromises temporal identity progression even in the presence of normal neighboring early-born neurons, where their feedback signal can proceed temporal pattern of NSC. Given that cell-cycle arrested NSCs have been clearly shown to proceed with their temporal pattern without their neuronal production owing to feedback of neurons from neighboring NSCs[10], the cell-autonomous effect of defective Fbl on temporal progression is unlikely due to just a defective cell cycle but rather likely through other pathways regulating the temporal pattern of NSCs.

**Fbl is essential for translation but not the transcription of epigenetic modifiers.** We investigated how Fbl affects temporal identity progression in NSCs. Considering that Fbl is an rRNA methyltransferase, we tested whether $Fbl$ knockout affects global protein synthesis by quantification of O-propargyl-puromycin (OPP) incorporation into nascent proteins (see "Methods"). We found decreased levels of newly synthesized proteins in DKO NSCs (Fig. 4). To investigate whether Fbl affects the translation of selected mRNAs, we performed ribosome profiling[34] and RNA-seq using $Fbl^{\Delta/+}$ and DKO brains (Fig. 5a). Translational efficiency (TE) can be calculated by comparing the levels of translating mRNA (Ribo-seq) and total mRNA (RNA-seq). After analyzing the quality of the data according to pre-established criteria (Supplementary Fig. 7a–c), we detected 299 and 541 genes with an increased and decreased TE ($q$-value $< 0.01$) in the brains of DKO, respectively (Fig. 5b, c, Supplementary Fig. 7d, and Supplementary Data 3). Given that $Fbl$ deletion reduced global levels of newly synthesized proteins, we considered that mRNAs with a lower TE in DKO could be regulated by Fbl. In fact, the Gene Ontology (GO) analysis showed that genes involved in

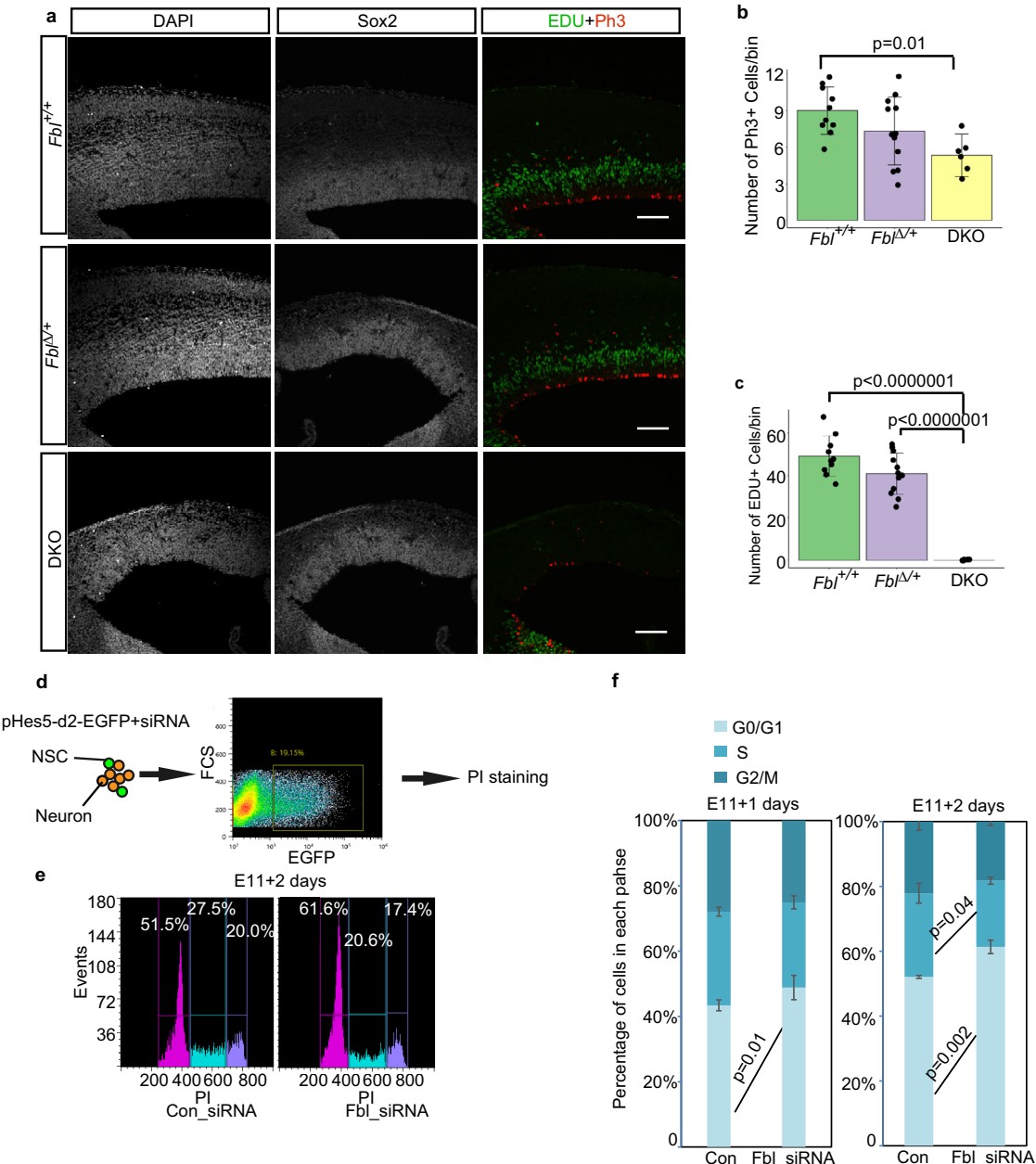

**Fig. 3 Fbl is essential for cell-cycle progression. a** Representative image of E14 brain sections stained for Sox2, Edu, and Ph3. Edu was injected 1 h before sampling. Scale bar: 100 μm. **b**, **c** Cell number quantification on sections based on immunostaining with the indicated markers (*n* = 5, 6, and 3 mice for Fbl + /+, FblΔ/+, and DKO, respectively; *n* = 2 sections per mouse; one-way ANOVA followed by Tukey's post hoc tests; data are presented as mean ± s.d. of counted sections). **d** Schematics of the experimental design to investigate neural stem cell (NSC) cell-cycle progression after treatment with control or Fbl siRNA. pHes5-d2-EGFP was used to label NSCs. To sort GFP + cells, the gate was chosen to ensure the negative controls did not include GFP + cells. **e** Cell-cycle analysis of NSCs after Fbl knockdown. **f** Proportion of G1/G0, S, and G2/M-phase change after Fbl knockdown for 1 day (left) or 2 days (right) (two-sided Student's *t* test; data are presented as mean ± s.d. of *n* = 3 biologically independent experiments). The gates of different cell-cycle phases were decided according to the boundary of different peaks. The gates were fixed in all experiments.

cell-cycle progression, such as *Cdk1* and *Cdk6*, were listed in the "centrosome" cluster as genes with a decrease in TE in DKO (Fig. 5c, Supplementary Fig. 7e, f, and Supplementary Data 3), and therefore cell-cycle defects could be attributed to a lower TE of those genes (Fig. 3). Interestingly, chromatin-related genes were highly enriched among those with decreased TE in DKO, consistent with the roles of epigenetic modifications in temporal patterning[35] (Fig. 5c and Supplementary Data 3). Among these, highly enriched genes were *Ezh2* and *Kdm6b*, which encode a methyltransferase and a demethylase, respectively, of H3K27me3 (Supplementary Fig. 7g, h). We found that the protein levels of

*Kdm6b* and *Ezh2* were reduced without a concurrent decrease in mRNA levels, while the protein and mRNA levels of the NSC markers Pax6 and Sox2 were not affected (Fig. 5d, e). This difference was not ascribed to protein stability (Supplementary Fig. 7i), thus we deduced that Fbl regulates protein translation.

To confirm the Fbl affects translation of *Ezh2* and *Kdm6b*, we also performed polysome profiling. Polysome profiling requires a large number of cells, but it is difficult to collect sufficient numbers of cells from embryonic brains. Therefore, after fractionating the samples, we pooled the fractions corresponding to the subpolysomes and polysomes and measured the expression

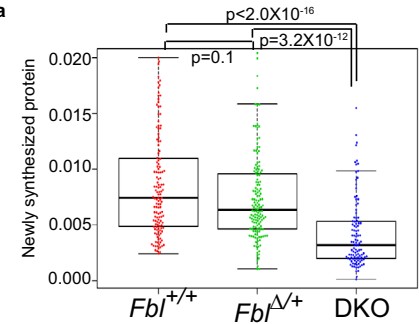

**a**

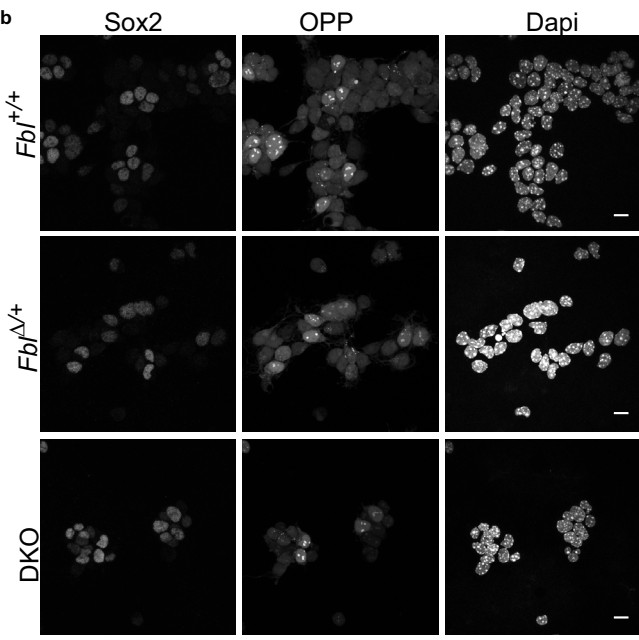

**b**

**Fig. 4 Decreased levels of newly synthesized proteins in DKO neural stem cells (NSCs). a** Quantification of O-propargyl-puromycin (OPP) incorporation in NSCs (Kruskal–Wallis test and Dunn's test with Bonferroni correction; Fbl + /+: $n = 135$; FblΔ/+: $n = 148$; DKO: $n = 106$). Data in the box–whiskers show maximum, third quartile to first quartile, and median and minimum. **b** Representative image of cultured cells stained for Sox2/ OPP/Dapi. Scale bar: 10 μm.

levels of mRNA in these pooled fractions. The results showed that when *Fbl* was knocked down, *Ezh2* mRNA was transferred from the polysome to the subpolysome, indicating that *Ezh2* translation was affected by the reduction of *Fbl* (Supplementary Fig. 7j), in accordance with the above results from ribosomal profiling. We note that *Kdm6b* mRNAs were not detectable due to their relatively low expression level in NSCs.

**Dynamics of genome-wide H3K4me3 and H3K27me3 distribution during temporal patterning of NSCs.** Because epigenetic regulator translations were affected by the deletion of *Fbl*, we focused on two major histone modifications: histone H3 trimethylation at Lys4 (H3K4me3) and H3K27me3, which is an active and repressive marker for gene expression, respectively. We performed H3K4me3 and H3K27me3 chromatin immunoprecipitation and sequencing (ChIP-seq) using fluorescence-activated cell sorting (FACS)-sorted NSCs from *Hes1-d2-EGFP* reporter mice from E11, E12, and E14 (Fig. 6a). We then analyzed how histone modification changes were associated with the temporal identity change of NSCs. We first analyzed the qualities of ChIP-seq data according to pre-established criteria (Supplementary

Fig. 8a–f). We subsequently classified each 200 bp chromosome region at each stage into one of four states based on two histone modification profiles: H3K4me3-only, H3K27me3-only, bivalent, and no-marker. The majority of genomic regions gained or lost H3K27me3 modification when shifting from E11 to E14. In contrast, changes in "H3K4me3-only" and "bivalent" regions were restricted to relatively small genomic regions (Fig. 6b). We extracted 1505 and 20 sites showing significantly differential intensities (*q*-value < 0.05) of H3K27me3 and H3K4me3 peaks, respectively, between E11 and E14 NSCs (Fig. 6c, d and Supplementary Data 4). Then, we focused on changes in those early- and late-onset genes as described above. The abundance of H3K27me3 peaks in early-onset genes showed a slight but significant difference between E11 and E14, and it decreased largely at E14 compared to E11 in late-onset genes (Fig. 6e, f), implying that H3K27me3 repressed the expression of late-onset genes in E11 NSCs. Conversely, the intensity of H3K4me3 peaks around the transcription start sites of early-onset genes had not drastically changed between E11 and E14 NSCs (Fig. 6g). In contrast, these peaks for late-onset genes increased from E11 to E14, reflecting the higher expression of these genes at E14 (Fig. 6h). Therefore, we concluded that dynamic changes in H3K27me3 deposition in early and late-onset genes, and in H3K4me3 deposition in late-onset genes, are associated with the temporal progression of NSC.

**Global H3K27me3 pattern predicted the developmental time of NSCs.** To test whether genome-wide H3K4me3 and H3K27me3 patterns could predict the developmental time of NSCs, we performed principal component analysis (PCA) based on the intensity of individual H3K27me3 and H3K4me3 peaks. The contribution of PC1-3 was more than other PCs (Supplementary Fig. 8g, h), we choose these PCs to visualize the data. H3K27me3 samples from different stages can be clearly separated and were deposited along time axes, while H3K4me3 samples were less distinguished especially for samples from E11 and E12 (Fig. 6i, j). As H3K27me3 patterns are highly related to the developmental stage, we conclude that H3K27me3 patterns within the genome can be considered as a part of the developmental clock in NSCs.

**GO term analysis of genes associated with H3K27me3 changes.** We then investigated what genes were associated with stage-specific H3K27me3 peaks. To this end, we extracted H3K27me3 peaks which showed significantly higher expression in E11 and E14, respectively (red points in Fig. 6c). Then, we searched for genes near these peaks using the GREAT[36], a bioinformatics tool. As a result, we identified 540 and 54 genes which were associated with H3K27me3 peaks at E11 and E14, respectively (Supplementary Data 5a, b). We observed many genes required for neuron differentiation and development are highly modified by H3K27me3 at E11 (Fig. 6k and Supplementary Data 5c). Repression of these genes by H3K27me3 is consistent with stage E11 when NSCs proliferate with suppression of neuronal differentiation. At E14, in contrast to E11, we observed genes that regulated the proliferation of NSCs were modified by H3K27me3, consistent with the neurogenic property of NSCs at this stage (Fig. 6l and Supplementary Data 5d).

**Deletion of *Fbl* affects H3K27me3 modification in NSCs.** We investigated histone modification changes upon *Fbl* deletion in DKO and control samples (including NSCs and progenies; Fig. 7a) and demonstrated alterations of H3K27me3 and H3K4me3 modifications at 669 and 0 sites (*q*-value < 0.05), respectively, indicating significant defects in H3K27me3 marks

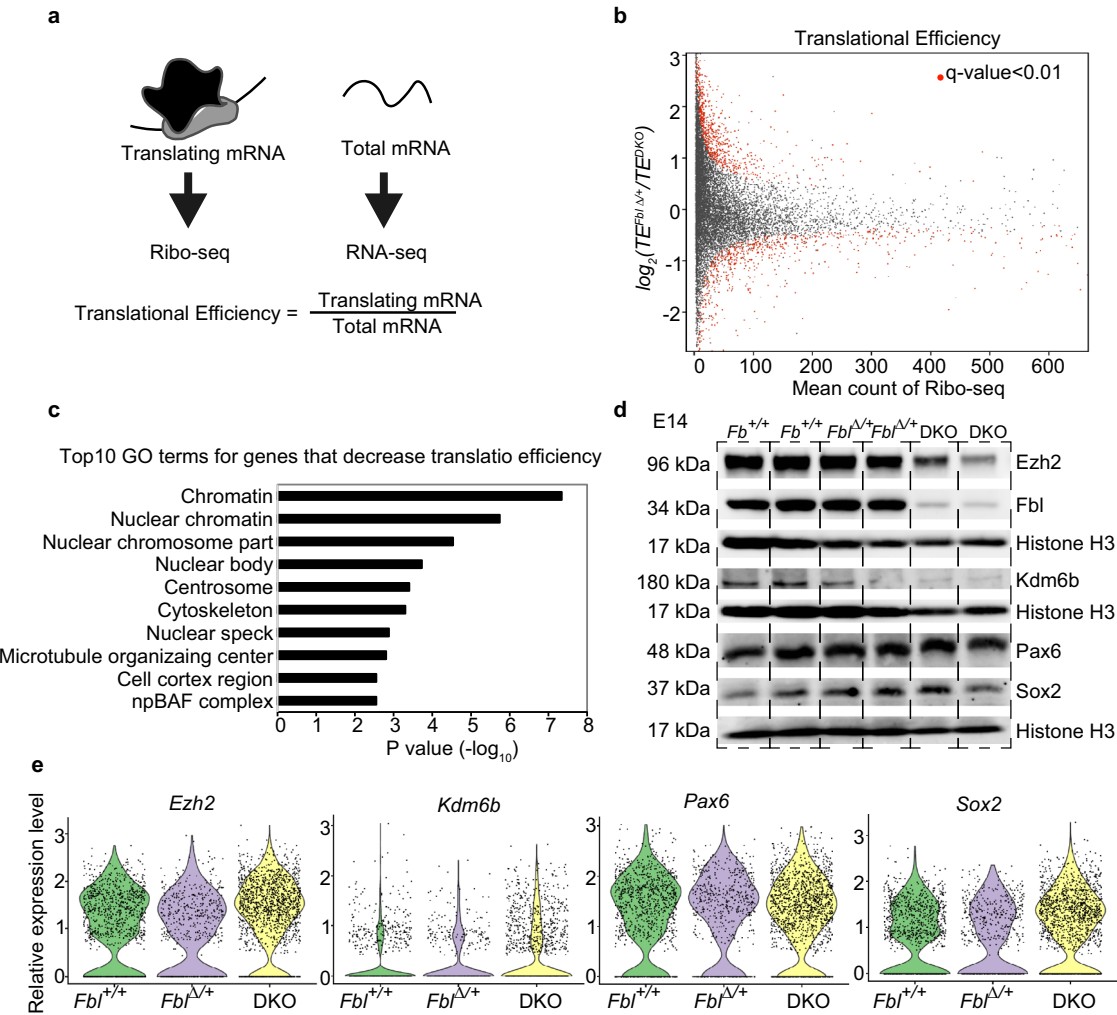

**Fig. 5 Fbl selectively regulates the translation of genes involved in H3K27me3 modification. a** Schematics of the experimental design for evaluation of translational efficiency (TE). **b** Mean-count and mean-difference plots comparing the observed and expected variance in TE. Genes with *q*-value < 0.01 are shown in red. **c** Top ten GO terms of transcripts showing reduced TE. **d**, **e** Western blotting (**d**) and single-cell RNA analysis (**e**) of the indicated genes, showing reduced protein levels, but not mRNA levels, of Ezh2 and Kdm6b in DKO brains at E14. Notice that Sox2 and Pax6 did not show changes in neither protein nor mRNA levels.

(Fig. 7b, c and Supplementary Data 6a). Furthermore, the average abundance of H3K27me3 peaks in early-onset genes showed a slight but significant difference between the control and DKO samples (Fig. 7d). Compared with control samples, the abundance of H3K27me3 peaks on late-onset genes (especially on transcription start sites) was clearly higher in DKO samples, indicating the expression of these genes was repressed by H3K27me3 (Fig. 7e). In fact, the expression of most of these genes (49/137) failed to upregulated in the NSCs of E14 DKO (Supplementary Data 6e). To investigate whether H3K27me3 defects in DKO can represent the temporal identity delay, we plotted these H3K27me3 samples together with previous samples (in Fig. 6j) using PCA. E14 DKO and control samples were deviated from E14 NSCs in the PCA plot due to contamination of differentiated progenitors. However, in terms of the temporal direction, the E14 DKO samples were not clustered with cells from E14 $Fbl^{+/+}$ nor $Fbl^{\Delta/+}$, but rather were closer to the E12 NSCs. This result reflected the delay of temporal identity progression of E14 DKO (Fig. 7f).

To confirm the role of H3K27me3 modification in temporal patterning, we collected cells from the E10 cortex and treated these cells with an inhibitor of methyltransferase: GSK-343

(2.5 µM) or a potential inhibitor of demethylase: GSK-J4 (0.5 µM) or both of these inhibitors for 3 days[37,38]. We then investigated gene expression changes after treatment with inhibitors by RNA-seq. Compared with control samples, the inhibition of H3K27me3 methyltransferase or demethylase alone affected the expression of 2366 and 117 DEGs, respectively (*q*-value < 0.05; Supplementary Data 6). To test the specificity of these inhibitors, we compared DEGs from this experiment with publicly available ChIP data and searched for factors that have target genes similar to DEGs. The results showed that in all cases (after GSK-J4 or GSK-343 treatment, or after double inhibition), Suz12, a subunit of the PRC2 complex responsible for H3K27me3, was the factor that has target genes most similar to the DEGs (Supplementary Fig. 9). These data suggest that our inhibitors specifically affect the modification of H3K27me3 at this concentration. We then calculated how the birthdate score was affected after the treatment of these inhibitors. The suppression of methyltransferase and demethylase was sufficient to reduce the birthdate score, while simultaneous inhibition of these two enzymes had more effect (Fig. 7g). Therefore, we concluded that both the writing and erasing of H3K27me3 are essential for temporal patterning and

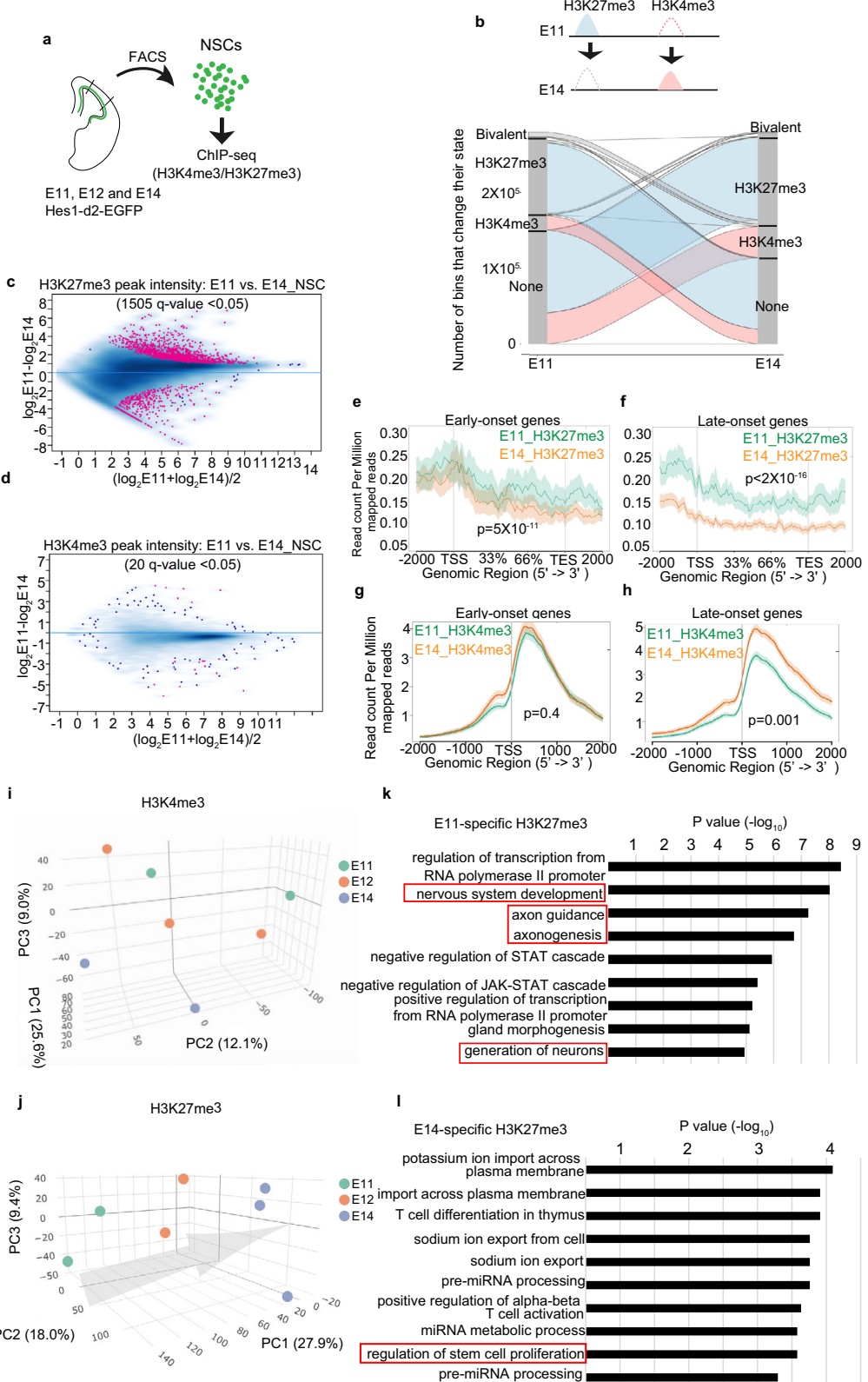

that Fbl facilitates these processes by controlling the translation of key enzymes.

**Characteristics of mRNAs affected by *Fbl* deletion.** Finally, we investigated the common feature of mRNAs affected by the deletion of *Fbl*. As the 5′-untranslated region (UTR) plays a critical role in translational regulation, we analyzed the 5′UTRs of

mRNAs showing downregulated TE after *Fbl* deletion. The minimum free energy (MFE) of the 5′UTRs of mRNAs showing downregulated TE by *Fbl* deletion was significantly lower than that of randomly selected mRNAs, suggesting that they were more structured (Fig. 8a). We also noted that 5′UTRs of some of these mRNAs tend to have a unique feature, namely, a poly (U) motif is highly enriched, while the significance of these motifs is

**Fig. 6 Genome-wide H3K4me3 and H3K27me3 modification change during temporal patterning of neural stem cells (NSCs). a** Experiment design for chromatin immunoprecipitation and sequencing (ChIP-seq). **b** H3K27me3 and H3K4me3 change of isolated Hes1+ NSCs from E11 to E14. Lines represent 200-bp chromosome regions. **c, d** Intensity comparison of H3K27me3 (**c**) and H3K4me3 (**d**) peaks between E11 and E14 NSCs, showing that H3K27me3 changed more dramatically than H3K4me3. Peaks with $q$-value < 0.05 are shown in red. **e–h** Read-density profiling of H3K27m3 (**e, f**) and H3K4m3 (**g, h**) at early-onset (**e, g**) and late-onset genes (**f, h**) in wild-type E11 and E14 NSCs. A two-sided Wilcoxon signed-rank test was used to evaluate the difference between E11 and E14 NSCs. The shaded areas represent standard errors. **i, j** Principal component analysis (PCA) of H3K27me3 and H3K4me3 peaks showing H3K27me3 samples can represent developmental time. Arrow indicates temporal axes. Different colored dots represent different stages of samples. **k, l** Top ten GO term of genes, with which stage-specific H3K27me3 peaks are associated.

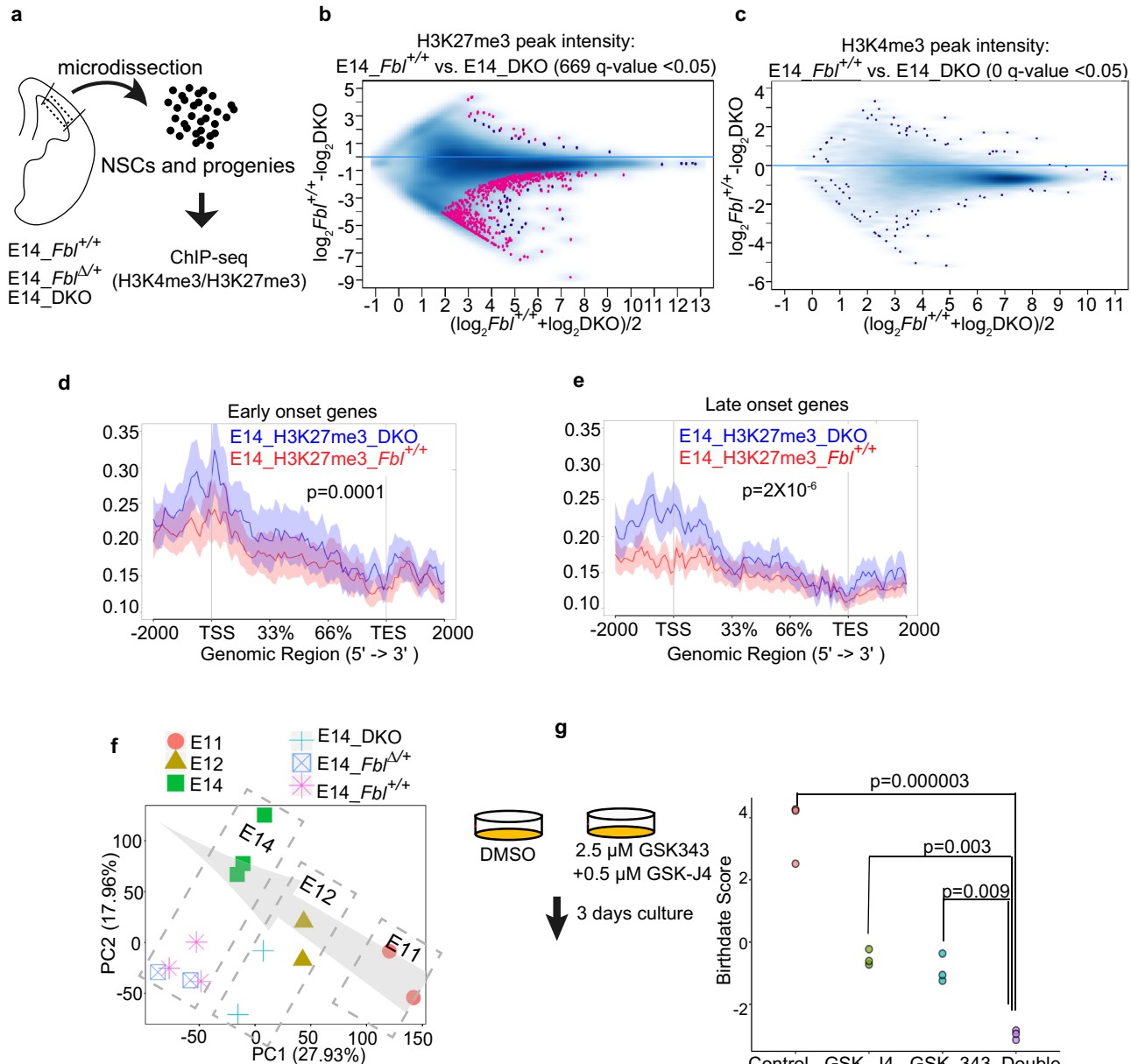

**Fig. 7 Fbl regulates the H3K27me3 pattern in neural stem cells (NSCs). a** Experiment design for chromatin immunoprecipitation and sequencing (ChIP-seq) using different genotypes. **b, c** Intensity comparison of H3K27me3 (**b**) and H3K4me3 (**c**) peaks between E14 control and DKO samples. Peaks with $q$-value < 0.05 are shown in red. **d, e** Read-density profiling of H3K27m3 at early-onset (**d**) and late-onset genes (**e**) in E14 control and DKO samples. The shaded areas represent standard errors. **f** Principal component analysis (PCA) of H3K27me3 peaks, showing DKO samples were located with E12 samples. Arrow indicates temporal axes. Different colored keys represent different stages or genotypes of samples. **g** Birthdate scoring after the inhibition of H3K27me3 methyltransferase, demethylase, and both of them, showing delayed temporal progression. Statistics analysis was performed by one-way ANOVA followed by Tukey's post hoc tests.

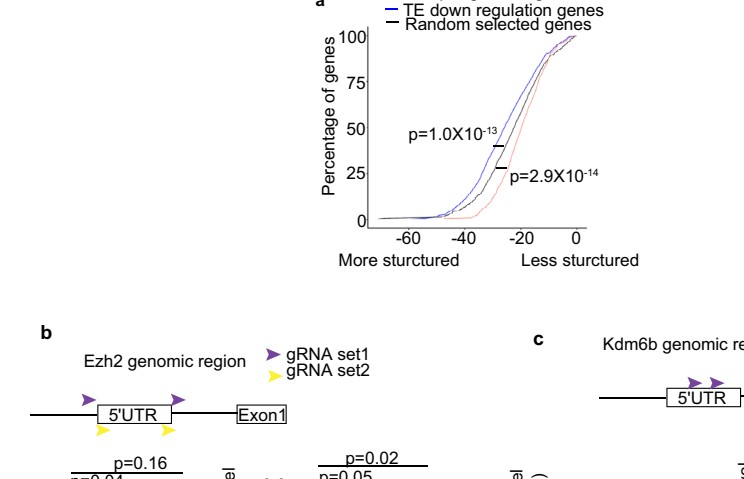

**Fig. 8 Fbl regulates translation through the 5′UTR in a cap-independent manner. a** 5′UTR minimum free energy (MFE) cumulative distribution of mRNAs showing changes in translational efficiency (TE) after Fbl knockout. Randomly selected mRNAs are shown as controls (two-sided Wilcoxon signed-rank test). **b, c** Changes in mRNA (left) and protein levels (right) after Ezh2 (**b**) and Kdm6b (**c**) 5′UTR knockout ($n = 3$; one-way ANOVA followed by Tukey's tests (**b**) and two-sided Student $t$ test (**c**); data are presented as mean ± s.d.).

unknown (31 genes; $P$ value = $10^{-56}$, Supplementary Fig. 10a). The characteristic of these 5′-UTRs may facilitate the translation of these mRNAs indirectly controlled by Fbl. Furthermore, to uncover the role of *Ezh2* and *Kdm6b* 5′-UTRs in vivo, we disrupted their 5′-UTRs in NSCs using CRISPR/Cas9 (Supplementary Fig. 10), resulting in significantly reduced protein levels, with mRNA levels either slightly or not affected (Fig. 8b, c and Supplementary Fig. 10b, c). These results suggest that 5′UTR of Ezh2 and Kdm6b is important for their translational regulation.

## Discussion

**Two types of developmental clock: cell-cycle-dependent and epigenetic regulator-dependent.** As a simple model to measure developmental time, the number of cell cycles can perfectly predict the initiation of transcription from the zygotic genome during early development of *Xenopus* (12 cycles) and *Drosophila* (10 cycles)[2,3]. However, in many cases, the developmental clock can work independently of the progression of the cell cycle. As described above, p27/kip1 accumulation during proliferation is important for the differentiation of oligodendrocyte precursors; however, slowing down the cell-cycle progression of oligodendrocyte precursors by cultures at 33 °C rather than 37 °C does not affect the differentiation process[39]. Moreover, during sequential expression of four temporal fate genes (*hunchback*, *Krüppel*, *pdm*, and *castor*) in *Drosophila* neuroblasts, the *hunchback* to *Krüppel* transition required cytokinesis, but the sequential expression of *Krüppel*, *pdm*, and *castor* is observed in the G2-arrest neuroblasts, indicating that their progression is independent of cell cycle[40].

DNA methylation-derived (5′methycytosine) epigenetic clock has been demonstrated to be highly related with chronological age[41]. However, the relationship between DNA methylation change and gene expression change during aging remains to be clarified. Chromatin modifications have been shown to be key switches in temporal patterning. The neurogenic-gliogenic potential shift of NSCs involves the function of polycomb repressive complex 2 in the development of the cerebral cortex.

However, a long-term and systematic operation of chromatin modification machineries in developmental progression, which can be described as a clock, has not been well established. In this study, we strongly suggested that H3K27me3 modifications operate as a developmental clock in the cerebral cortex development. We showed that the patterning of H3K27me3 peaks is highly related to the progression of the developmental stage (Figs. 6j and 9). Moreover, we observed that the initiation of expression of late-onset genes is accompanied by the release of H3K27me3 marker from their gene body (Fig. 6f). These results suggested that at least in part, the H3K27me3-dependent mechanism directly controls gene expression, promoting the fate transition in NSCs in a clock-like way across the genome. In contrast to H3K27me3, H3K4me3 modification is less changed with developmental stages. We observed both H3K4me3 and H3K27me3 peaks in the genomic region of early-onset genes (Fig. 6e, g). This bivalent state is considered a mechanism for maintaining the potential for genetic activation[42]. This bivalent state of early-onset genes may also explain the plasticity of late NSCs, in which reprogramming of temporal identity has been reported after transplantation into young brains[43].

**The function of Fbl in temporal fate transitions is independent of cell cycle and differentiation.** Deletion of *Fbl* affects both differentiation and temporal progression in NSCs (Fig. 2e, f). Importantly, the differentiation process is independent of the temporal progression of NSCs. For example, overexpression of NICD is enough to repress the differentiation process in NSCs, however, it is not sufficient to compromise the temporal fate progressions. Once the NICD is removed, NSCs resume generating the same neurons as those without temporary inhibition of neurogenesis by Notch activation[44]. Thus, differentiation failure observed in *Fbl* mutants should be considered separately with temporal patterning defects.

We have also demonstrated that Fbl regulates the temporal fate transitions of NSCs independent of the cell cycle, and even Fbl

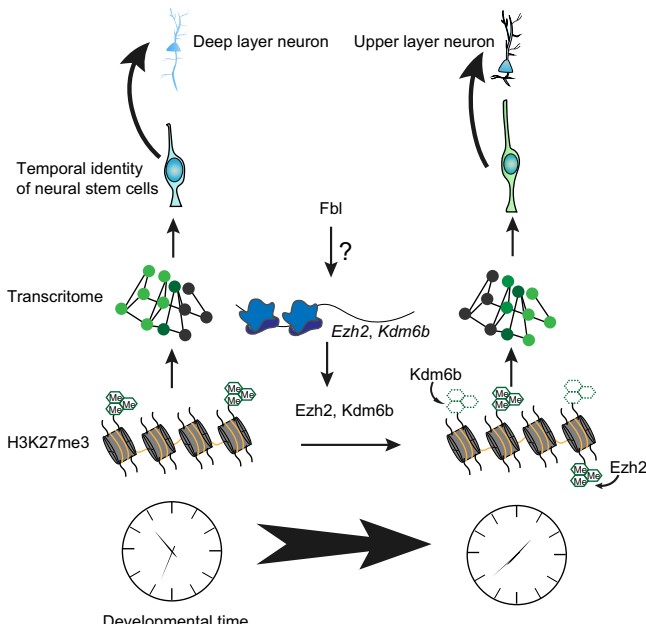

**Fig. 9 Fbl drives the developmental clock of neural stem cells (NSCs).** Fbl enhances the translation of Ezh2 and Kdm6b. Ezh2 and Kdm6b change H3K27me3 pattern in NSCs. H3K27me3 patterning further affects gene expression change and regulates the temporal fate of NSCs. However, it is still unknown how Fbl affects the translation of Ezh2 and Kdm6b.

plays an essential role in the regulation of S-phase initiation. The functions of Fbl in cell-cycle progression are also observed in zebrafish[24]. Unlike the mouse, in which straight knockout of *Fbl* lead to embryonic lethal at very early stages (around E2)[23], *Fbl*-null mutant zebrafish survives to later developmental stage. These mutant fish showed a defect of S-phase progression in the dorsal midbrain and retina, but the ventral region of the brains was less affected[24]. It is interesting to investigate the function of the Fbl-mediated developmental clock in zebrafish.

**Possible mechanisms for Fbl to regulate translation.** The post-transcription regulation of Ezh2 by microRNA is also involved in the differentiation of adult NSCs in the dentate gyrus of the hippocampus, the maturation of the neuron in the hippocampus, and the fate choice between neuronal and astrocyte differentiation in embryonic NSCs[45,46]. Taken together with our results showing the translational regulation of Ezh2, we conclude that precise control of the expression of Ezh2 is important for NSCs. While microRNA degrades the mRNA level of *Ezh2* by directly target its 3′-UTR, it is unclear how Fbl regulates the translation of *Ezh2* and *Kdm6b*. In human cells, Fbl affects rRNA methylation and controls mRNA translation through internal ribosome entry site (IRES) elements on their 5′UTR[22,25]. The systematic screening of 5′UTR of human genes suggests that some genes maintain the structure of IRES. They found that IRES elements are characterized by the presence of poly (U) motif or a highly organized secondary structure[47]. These characteristics were also observed in the mRNAs that downregulated their translational efficiency in the absence of Fbl in this study. Recently, a study reported that Ezh2 binds directly to Fbl, affects 2′-O-methylation in rRNA, and impacts the translation, of oncogenes through IRES through their 5′UTR in cancer cells[48]. However, there are many technological limitations to the usage of bicistronics to evaluate the activity of cellular IRES[49]. Therefore, in the future, it should be carefully investigated whether Fbl regulates the translation of *Ezh2/Kdm6b* in NSCs through IRES.

Fbl also affects histone H2A methylation and regulates the transcription of rDNA into rRNA[50]. Thus, Fbl may affect global protein synthesis through reduction of rRNA. Recently, 2′-O-methylation sites were found in mRNAs[51,52] where they inhibit translation elongation by slowing tRNA decoding[53] rather than by facilitating translation as found in this study. Therefore, it is more likely that modifications of rRNA, rather than mRNA, mediate the effect of Fbl on translation during temporal patterning in the brain. However, we did not have direct evidence to show that Fbl modification of rRNA facilitates ribosomes to recognize or bind 5′UTR of target genes. The development of specific inhibitors for 2′-O-methylation of rRNA may help answer this question.

Why does epigenome-mediated temporal patterning utilize translational control by Fbl as an upstream mechanism to advance the developmental clock, in addition to transcriptional control of epigenetic factors? We speculated that selective translational promotion is a simple and efficient method to ensure the production of specific protein groups; the transcriptional upregulation of a group of genes often requires the cooperation of many epigenetic and transcription factors. Translational control of epigenetic modifiers by Fbl adds another level of complexity to gene expression, and thus, greatly widens the range of the epigenetic landscape, notably impacting diverse developmental processes and diseases.

## Methods

**Weighted gene coexpression network analysis (WGCNA).** WGCNA was performed using microarray data from E11 ($n = 24$) and E14 ($n = 31$) single cells[10]. We used top 10,000 genes after ranking of all genes by their expression level. Soft power parameter was set at three and dpSplt parameter was set as 0. Genes in the brown module were used to analyze protein interaction (https://string-db.org) and top-10 nodes were identified by cytoHubba in Cytoscape[54]. Gene enrichment analysis was performed using Enrichr[55].

**Animals.** All animal procedures were performed in accordance with the guidelines for animal experiments at RIKEN Center for Biosystems Dynamics Research.

**Mice.** To produce conditional knockout mice of *Fbl* (Accession No. CDB0137E: http://www2.clst.riken.jp/arg/mutant%20mice%20list.html), *loxP* sequences were introduced on both sides of *Fbl* locus by genome editing technology using CRISPR/ CAS9 system in mouse zygotes. crRNA(CRISPR RNA), tracrRNA(trans-activating crRNA) and donor single-stranded oligodeoxynucleotides (ssODNs) consisting of a *loxP* site, an EcoRV recognition site and homology arms were chemically synthesized (Fasmac): 5′-crRNA (5′-AGCUUGUCUCAGGUUUAACCGUUUUAGA GCUAUGCUGUUUUG-3′), 3′-crRNA (5′-UCAAGGGCGCAUGCGUCUCGGU UUUAGAGCUAUGCUGUUUUG-3′) and tracrRNA (5′-AAACAGCAUAGCAA GUUAAAAUAAGGCUAGUCCGUUAUCAACUUGAAA AAGUGGCACCGAG UCGGUGCU-3′), 5′-loxP ssODN (5′-GTCCTCAGCACACAGCTTGTCTCAGGT TTAGATATCATAACTTCGTATAGCATACATTATACGAAGTTATACCTGG TTCCACATCACACCTGCCGCTGTT-3′) and 3′-loxP ssODN (5′-CACACAAA GTTGATCAAGGGCGCATGCGTCATAACTTCGTATAATGTATGCTATACG AAGTTATGATATCTCGAGGCCACTTAGCAATAGGCACCAGACA-3′). The mixture of ssODNs, crRNAs, tracrRNA, and Cas9 protein was injected into pronuclei of C57BL/6N zygotes by microinjection, and the injected zygotes were transferred into the oviducts of pseudopregnant ICR female mice. The resultant offspring were genotyped by genetic PCR with combination of following primers: 5′-loxP site: forward: 5′-CTCTTCTAGGACACTCCATCCCTTATCAAG-3′; reverse: 5′-AGTACTAGTTGTGAAGGTATGAGAGGGGTC-3′; (wild type: 489 bp and 5′loxP: 529 bp) 3′-loxP site: forward: 5′-GAAGAAGATGCAGCA GGAGAACATGAAGCC-3′; reverse: 5′-CAACCAGCAAAATGGCGACCACAA-CAAACC-3′ (wild type: 575 bp and 3′loxP: 615 bp). The insertion of the *loxP* site was confirmed by the EcoRV digestion (5′-loxP: 292 bp and 237 bp, and 3′-loxP: 360 bp and 255 bp) and sequencing. The germline transmission of the floxed allele in which 5′- and 3′-loxP sites were in the same allele was confirmed by crossing with C57BL/6. The breeding of *Trp53* mutant mice was described previously[56]. *Fbl* and *Trp53* mutant mice were maintained in C57BL/6 background. The reporter mouse line: *pHes1–d2-EGFP* (a gift from R. Kageyama[26,57]) was maintained in an ICR background. Wild-type mice used for inhibitors treatment were maintained in ICR background.

**Immunohistology and confocal imaging.** Brains were fixed in 1% paraformaldehyde (PFA) overnight, treated with 25% sucrose for cryoprotection, and

then embedded in OCT compound (Tissue-Tek; Sakura). Sections (12 μm) were made using a cryostat (CM3050S Leica Microsystems). For immunostaining, sections were blocked with 3% skim milk powder in PBST (0.1% Tween20 in PBS) for 1 h at room temperature (RT), followed by incubation with primary antibody at the optimized concentration at 4 °C. According to primary antibodies, sections were then incubated with secondary antibodies with labeled fluorescent probes (1:400) (Alexa Fluor 488, cy3, or 647; Jackson ImmunoResearch) for 90 min at RT. DAPI was used for nuclei detection. A scanning confocal microscope (Olympus FV1000 or Zeiss LSM 880 with Airyscan) was used for observation. For Tbr2/EOMES staining, the antibody (1:400, rat monoclonal, clone Dan11mag; eBioscience at Thermo Fisher) has been conjugated with eFluor660. The primary antibodies were: fibrillarin (1:1000, rabbit polyclonal, ab5821, Abcam), Cleaved Caspase-3 (Asp175) (1:400, rabbit polyclonal, 9661S, Cell Signaling Technology), Satb2 (1:400, mouse monoclonal, ab51502, Abcam), Tbr1 (1:400, rabbit polyclonal, ab31940, Abcam), Olig2 (1:400, goat polyclonal, AF2418, R&D System), Pax6 (1:400, rabbit polyclonal, PRB-278P, Covance), Sox2 (1:400, goat polyclonal, sc-17320, Santa Cruz), GFP (1:400, chick polyclonal, GFP-1020, Aves), Phospho-Histone H3 (Ser10) (1:400, rabbit polyclonal, 06-570, Millipore), Brn2 (1:400, goat polyclonal, sc-6029, Santa Cruz), FoxP2 (1:400, goat polyclonal, sc-21069,, Santa Cruz), Zbtb20 (1:400, rabbit polyclonal, HPA016815, Sigma) and Dmrt3 (1:1000, rabbit polyclonal, a gift from D. Konno)[58].

**Western blotting analysis.** Pierce® IP lysis buffer (Thermofisher Scientific) was added into the collected dorsal cortices or the FACS-sorted cells. After sonication (TOMY HandySonic, 10 s at level 4), the lysates were centrifuged at 13,000×g for 10 min. The supernatant was mixed with 1 μl of protease inhibitor cocktail (Nicolai Tesque). After mixing with the same volume of sampling buffer Laemmli (Sigma), the supernatant was boiled at 98 °C for 5 min, applied to SDS page gel (SuperSep, WAKO), and transferred onto a 0.2 μM nitrocellulose blotting membrane (GE Healthcare). After incubation with blocking buffer (5% milk in tris-buffered saline with 0.1% Tween20) at RT for 1 h, the membrane was incubated with primary antibody overnight at 4 °C, followed by the incubation with anti-rabbit or anti-mouse IgG antibody conjugated to horseradish peroxidase (NA934V or NA931V, GE Healthcare). After reactivation with Chemi-Lumi One Ultra (Nacalai Tesque), images were obtained using the LAS3000 mini imaging system (Fujifilm). The intensity of the bands was calculated using ImageJ (1.52d). The primary antibodies were: Ezh2 (1:1000, mouse monoclonal, 5246S, Cell Signaling Technology), Pax6 (1:1000, rabbit polyclonal, PRB-278P, Covance), Histone H3 (1:1000, rabbit monoclonal, 4499, Cell Signaling Technology), Kdm6b (1:1000, rabbit polyclonal, NBP1-06640, Novus Biologicals), Sox2 (1:1000, rabbit polyclonal, ab75179, Abcam), Fibrillarin (1:1000, rabbit polyclonal, ab5821, Abcam) and α-tubulin (1:1000, mouse monoclonal, clone DM1A, T9026, Sigma-Aldrich).

**Plasmid, stealth siRNA, and gRNA.** Stealth siRNA for *Fbl* and control was designed with BLOCK-iT™ RNAi Designer (https://rnaidesigner.thermofisher.com/rnaiexpress/) as follows: *Fbl* siRNA: 5′-CCGCAUCGUCAUGAAGGUGUCUUUA-3′ and control siRNA: 5′-CCGGCUACUAGUGGAUGUUCCAUUA-3′ (Thermofisher).

The *pHes5-d2-EGFP* plasmid used in the cell-cycle analysis was a gift from R. Kageyama[26,57].

The target region of gRNA for knockout of *Fbl* and *Trp53* is shown below: *Fbl* gRNA1: 5′-CCACCATGCGGCATGCTGGAATT-3′; *Fbl* gRNA2: 5′- CCTCGAGACGCATGCGCCCTTGA-3′; *Trp53* gRNA1: 5′-CCTCGCATAAGTTTCCTGAAATA-3′; *Trp53* gRNA2: 5′-CAGCAGGTGTGCCGAACAGGTGG-3′.

The target region of gRNA for knockout of 5′UTR of *Ezh2* and *Kdm6b* is shown below:

*Ezh2* gRNA set1_1: 5′-GGGTTGCTGCGTTTGGCGCTCGG-3′; set1_2: 5′-CCGTCGGCCGCCGGTGGTCGGCA-3′; *Ezh2* gRNA set2_1: 5′-CCGGTCGCGTCCGACACCCAGTG-3′; set2_2: 5′-GAGAGGCGGGCTGGCGCGCGG-3′; *Kdm6b* gRNA set1_1:5′-CCCTCAGGTCGGCTCGTGAATGG-3′ set1_2: 5′-CCCACTTGCGCGATTCTAGGGGC-3′. Primers for the production of gRNA were designed following[59]. Designed primers were self-amplified by PCR and PCR fragments were inserted into AflII cut gRNA vectors modified from Church Lab[60].

All plasmids were purified with endotoxin-free NucleoBond Xtra Midi EF kit (Macherey–Nagel).

**Calculation of rRNA level.** Dorsal cortices were removed from E14 mice with different genotypes. RNA extraction was performed using the Qiagen RNeasy kit following the manual. To examine the level of rRNA, we used a previously published method with modification[21]. This method allows us to estimate the degree of methylation of a particular position in rRNA, based on the fact that reverse transcription reaction will be inhibited in conditions of low dNTP concentration, but not in conditions of high dNTP concentration. Briefly, 10 ng of RNA was mixed with 1 μM random primer and 1 mM dNTP (high condition) or 0.004 mM dNTP (low high) and incubated at 65 °C for 5 min. Then, 5× First Stand buffer (Invitrogen), 10× SuperScript® III Reverse Transcriptase (Invitrogen), 0.005 mM DTT and 20× RNase inhibitor (TAKARA BIO) were added and the mix was incubated at 50 °C for 1 h followed by 70 °C for 15 min. QPCR was then performed to determine the dosage of the amplicon of each primer at different conditions with

ReverTra Ace qPCR Master Mix (TOYOBO, FSQ-201) in a Thermal Cycler Dice Real-Time System Single (TaKaRa, TP850) with TB Green Premix Ex Taq II (Tli RNaseH Plus) (TaKaRa, RR820). Following primer sets were used: 28S_1673 forward: 5′-CTAGTGGGCCACTTTTGGTA-3′; 28S_1673 reverse: 5′-TTCATCCCGCAGCGCCAGTT-3′; 28S_2614 forward: 5′-TAGGTAAGG-GAAGTCGGCAA-3′; 28S_2614 reverse: 5′-CCTTATCCCGAAGTTACGGA-3′; 28S_3441 forward: 5′-ATGACTCTCTTAAGGTAGCC-3′; 28S_3441 reverse: 5′-TCACTAATTAGATGACGAGG-3′; 28S_4223 forward: 5′- GGTTAGTTT-TACCCTACTGA-3′; 28S_4223 reverse: 5′-GATTACCATGGCAACAACAC-′3; 28S_4242 forward: 5′-TGATGTGTTGTTGCCATGGT-3′; 28S_4242 reverse: 5′-GTTCCTCTCGTACTGAGCAG-3′; 28S_3958 forward: 5′-CTCGCTTGATCTT-GATTTTC-3′; 28S_3958 reverse: 5′-CGCTTTCACGGTCTGTATTC-3′; 28S_4188 forward: 5′-TAGGGAACGTGAGCTGGGTTTAGA-3′; 28S_4188 reverse: 5′-GTAAAACTAACCTGTCTCACGACG-3′; Methylation level of rRNA was calculated as dosage of amplicon at low dosage of dNTP of amplicon at high dosage of dNTP.

**EDU staining.** Pregnant mice were injected intraperitoneally with EdU (Invitrogen) at 12.5 mg/kg. EdU staining was performed with the Click-iT EdU Imaging kit (Invitrogen).

**Fbl knockdown and inhibitors treatment of brain cells.** For knockdown experiments, cells from E10 dorsal cortices were counted and resuspended with buffer R (Neon™ transfer system, Invitrogen) in the concentration of $8 \times 10^6$ cells/mL. Resuspended cells (100 μL) were mixed with 160 μM control or *Fbl* siRNA. Electroporation was performed with Neon™ transfer system, Invitrogen) at conditions of 1600 voltage, 20 width and 1 pulse. These transfected cells were mixed with 2 ml culture medium (20 ng/ml human basic FGF (Peprotech), 1XB27 RA- (Gibco), in Dulbecco's Modified Eagle Medium (DMEM) F12 + GlutaMax (Gibco) and distributed into 4-well or 24-well plates (500 μL/well) and cultured at 37 °C. For inhibitors treatment, $2 \times 10^5$ cells/well were culture in the DMSO, 0.5 μM GSK-J4 (Sigma, SML0701), 2.5 μM GSK-343 (Sigma, SML0766), or 0.5 μM GSK-J4 and 2.5 μM GSK-343 for 3 days.

**Cell-cycle analysis.** For cell-cycle analysis, *Hes5-d2-EGFP* plasmid (2 μg) with 160 μM control or Fbl siRNA was transfected into resuspended cells from E11 dorsal cortices as mentioned above. After 2 days, cells were harvested for sorting; The GFP-positive NSCs were sorted into 1 mL of 0.375% BSA/PBS with SH800 (SONY). Then, ice-cold 100% ethanol (3 ml) was added into these sorted cells with a gentle vortex. These cells were fixed at −30 °C for 1 h, followed by centrifugation at 1500×g for 5 min. After removal of the supernatant, 700 μL of staining solution (50 μg/mL propidium iodide (Nicolai Tesque) and 100 μg/mL RNase A in 1% BSA/PBS) was added and mixed well by pipetting. The cell sorting was performed by SH800 according to the manufacturer's protocol. The gates of different cell-cycle phases were decided according to the boundary of different peaks. The gates were fixed in all experiments.

**Knockout of *Fbl* and *Trp53* by CRISPER/Cas9 system.** The gRNA plasmid together with pCAX-Cas9 were transfected into cells from E10 dorsal cortices with NEON (Neon™ transfer system, Invitrogen) at the condition of 1600 voltage, 20 width, and 1 pulse. To label knockout cells, EGFP was simultaneously knocked into beta-actin locus as described previously[59]. For $2 \times 10^5$ cells, 0.5 μg of each plasmid was used. For clone analysis, 10,000 electroporated cells were mixed with 190,000 untreated cells and cultured for 4 days. The culture medium was changed every 2 days.

To knockout of *Fbl* and *Trp53* in utero, 0.3 ug/μL gRNA sets and pCAX-Cas9 were electroporated into dorsal brains at E11. In utero electroporation in mice was reported previously[61,62].

**Single-cell isolation and library construction.** To isolate *Hes1* positive neural stem cells (NSCs) at E14, dorsal cortices were dissected from *Hes1-d2-EGFP*$^{Tg/+}$ mice. Cortices were dissociated with 0.05% trypsin with Hanks' balanced salt solution (HBSS) (−) at 37 °C for 10 min. After centrifugation at 1000×g for 5 min, cells were resuspended with 0.375% BSA/HBSS(−) by gentle pipetting 15 to 20 times. Resuspended cells were filtered with 35-μm filter (Falcon) and sorted into sorting buffer (20 ng/ml human basic FGF (Peprotech), 1×B27 RA- (Gibco), in Dulbecco's Modified Eagle Medium (DMEM) F12 + GlutaMax (Gibco)) by a cell sorter (SH800, SONY) equipped with 130-μm sorting chips (SONY, LE-C3113). After sorting, cell number was counted by Countess or Countess II (Invitrogen). For cells from wild-type and *Fbl*-mutant mice, cells were collected similarly except that sorting was not performed.

Collected cells were immediately loaded into the 10X-Genomics Chromium (10X Genomics, Pleasanton, CA). Libraries for single-cell cDNA were prepared using Chromium 3′ v2 platform as the manufacturer's protocol.

**Bioinformatics analysis of single-cell data.** Sequenced data was mapped to mm10 and cell number and raw count for each gene were reported by Cellranger 2.0.2 (10X GENOMICS). All data were further analyzed using the bioinformatics

pipeline Seurat (2.3.4)[63]. Briefly, we first created a SeuratObject, in which genes that were expressed by less than three cells and cells that expressed less than 200 genes were removed. The unique molecular index (UMI) was automatically counted by Seurat. We calculated the percentage of UMI mapping to mitochondrial genes and used these values to further filter cells (<15%). Data were then normalized ((normalization.method = "LogNormalize", scale.factor = 100,000) and scaled (vars.to.repress = c("UMI","percent.mito")). Then, principal component (PC) analysis was performed and the top-10 PCs were used for t-Distributed Stochastic Neighbor Embedding (t-SNE) dimensional reduction. TSNEPlot and FindAllMarkers were used to visualize clusters and to investigate the cell type of each cluster, respectively.

**Pseudotemporal ordering.** To order NSCs pseudotemporally, the monocle (2.10.1) package was used[64]. The NSCs were extracted based on t-SNE clustering (Cluster 0, 1, 5, 7, 11, 12 in Supplementary Fig. 5b). The NSCs were manually clustered into early NSCs and late NSCs according to the expression level of *Hmga2* (>2) and *Dbi* (>15). Then, these cells were ordered by DDRTree method according to 200 differential expression genes between these two clusters.

**Calculation of birthdate and differentiation score.** To evaluate the temporal identity and differentiation state of each cell, we used core genes to calculate the birthdate and differentiation score as previously reported[12]. Briefly, the authors used ordinal regression models to predict the birthdate and differentiation state of each cell. The best 100 genes were selected based on the linear weight of the models for prediction. We used 95 genes (5 of them could not be detected in our system, including Rp23-379c24.2, Leprel1, Rp23-14p23.9, Mir99ahg and Yam1) and 100 genes, respectively, for calculation of birthdate and differentiation score by the following formula:

$$\text{Birthdate score of the cell } j = \sum_{i=1}^{95} \mathbf{WB_i} * \mathbf{EXP_{ij}} \tag{1}$$

$$\text{Differentiation score of the cell } j = \sum_{i=1}^{100} \mathbf{WD_i} * \mathbf{EXP_{ij}} \tag{2}$$

$\mathbf{WB_i}$, $\mathbf{WD_i}$, and $\mathbf{EXP_{ij}}$ indicates weight of each temporal-related gene, differentiation-related gene ($i$) and its expression level in the cell j, respectively.

**Ribosome profiling.** Ribosome profiling protocol was modified from previous study[65]. Briefly, E14 dorsal cortices were dissected and removed into a 1.5 ml tube. Then, 200 μL and 100 μL (in the case of $Fbl^{\Delta/+}$ and DKO cortices, respectively) ice-cold lysis buffer (20 mM Tris-HCl (pH 7.5), 150 mM NaCl, 5 mM MgCl₂, 1 mM DTT, 100 μg/ml cycloheximide, and 1% Triton X-100) was added. The tissues were lysed by pipetting and these lysates were incubated with 15 U of TURBO DNase (Invitrogen) for 10 min on ice. The supernatants were collected after centrifugation at 20,000 × g for 10 min at 4 °C. The Qubit RNA HS Assay Kit (Thermo Fisher Scientific) was used to measure the RNA concentration. Supernatants containing 400 ng RNA were treated with 0.8 U of RNase I for 45 min at 25 °C. Then, the same process was performed as previously described[65]. To remove rRNAs from the total RNA, the Ribo-Zero Gold rRNA Removal Kit (Human/Mouse/Rat) (Illumina) was used.

**Ribo-seq and RNA-seq analysis.** Sequence data from RNA-seq and ribosome profiling were trimmed with Trim Galore! (-phred33 -q 30 -length 35) (https://www.bioinformatics.babraham.ac.uk/projects/trim_galore/). Cutadapt (https://cutadapt.readthedocs.io/en/stable/guide.html) was used to remove universal adaptors and linker sequences. Reads were then mapped onto mouse genome mm10 using hisat2 (2.1.0)[66] and rRNA was removed from mapped reads. Duplicates were marked and removed with Picard (https://broadinstitute.github.io/picard/). For Ribo-seq, Ribodiff[67] was used to count reads and genes having more than 10 counts in ribo-seq (11370 genes) were used to detect genes with different translational efficiency. For RNA-seq, Stringtie (1.3.6)[68] was used to identify differentially expressed genes between experiments and the result was further analyzed by TCC (1.22.1)[69].

**Chromatin immunoprecipitation (ChIP) analysis.** Cells were either FACS-sorted from E11, E12 and E14 Hes1-d2-EGFP[TG/+] mice or from E14 cortices of control ($P53^{-/-}$ $Emx1^{Cre/+}$, $Fbl^{flox/+}P53^{-/-}$ and $Fbl^{flox/flox}P53^{-/-}$), heterozygotes ($Fbl^{flox/+}P53^{-/-}$ $Emx1^{Cre/+}$) and DKO ($Fbl^{flox/flox}P53^{-/-}Emx1^{Cre/+}$). The number of cells was counted by Countess or Countess II (Invitrogen). These cells were fixed with 0.25% PFA in PBS for 10 min at RT, washed with 0.1 M glycine in PBS three times. These cells were collected after centrifugation at 1500×g for 5 min. After removal of the supernatant, the cells were resuspended with ChIP buffer (10 mM Tris-HCl pH 8.0, 200 mM KCl, 1 mM CaCl₂, 0.5% NP40) at a concentration of 1 × 10⁶ cells/mL. After a brief sonication (TOMY HandySonic, 10 s, level 10), a micrococcal nuclease (Worthington) was added at the concentration of 50 U/ml. The mixer was incubated at 37 °C for 20 min. EDTA (final concentration: 10 mM) was added to stop MNase reaction. The lysates were collected after centrifugation at 15,000 × g for 5 min and supernatants were removed. The lysates were resuspended with RIPA buffer (50 mM Tris-HCl pH 8.0, 150 mM NaCl, 2 mM EDTA, 1% NP40, 0.5% sodium deoxycholate, 0.1% SDS). After sonication for three times (10 s, level 10), lysates were centrifuged at

15,000 × g for 5 min and supernatants were collected. For each ChIP experiment, 100 μL lysate was used. In total, 25 μL dynabeads-anti-rabbit or mouse IgG (Invitrogen) were washed with 500 μL ChIP buffer. The beads were incubated with 1 μL of primary antibody H3K27me3 (rabbit monoclonal, Cell Signaling Technology, #9733) and H3K4me3 (monocle, Wako, 307-34813) in 300 μL blocking buffer (5 mg/ml BSA, 0.5% NP40, 0.1% Tween20 in PBS) at 4 °C with gentle rotation overnight. After washing with ChIP buffer three times, those beads were mixed with 400 μL blocking buffer and 100 μL ChIP lysate and incubated for 1 h at 4 °C. These beads were washed five times with low salt wash buffer (20 mM Tris-HCl pH 8.0, 150 mM NaCl, 2 mM EDTA, 1% Triton-X-100, and 0.1% SDS) and high salt wash buffer (20 mM Tris-HCl pH 8.0, 500 mM NaCl, 2 mM EDTA, 1% Triton-X-100 and 0.1% SDS), respectively, followed by the release of chromatin by incubation of these beads with 200 μL elution buffer (50 mM Tris-HCl pH 8.0, 10 mM EDTA and 1% SDS) for 30 min at 65 °C. The supernatant was removed into a new tube and was incubated at 65 °C for 4 h. 1 μL RNase A (Sigma) was added and incubated at 37 °C for 10 min to degrade RNA. The supernatant was incubated at 55 °C overnight with 5 μL proteinase K (Roche). Genomic DNA was extracted by phenol:chloroform extraction.

**Bioinformatics analysis of ChIP data.** Sequenced reads with poor quality were trimmed with Trim Galore! (-phred33 -q 30 -length 35) (https://www.bioinformatics.babraham.ac.uk/projects/trim_galore/). Reads were mapped onto mouse genome mm10 using bowtie[70] with the parameter –m1 -best -strata. Mapped sam files were transferred into bam files and were sorted with samtools (1.5)[71]. Duplicates were marked and removed with Picard (http://broadinstitute.github.io/picard/). Peaks of the ChIP-seq were called using MACS2 (2.1.1)[72]. The Q-value for the cutoff of H3K4me3 peaks was set at 0.01. For the call peaks of H3K27me3, a broad function was used and the q-value was set at 0.01 and 0.05 to cutoff narrow/strong regions or broad/weak regions, respectively. For each sample, specified input was used as a control.

Deeptools (3.2.1)[73] was used to calculation the correlation of each dataset. The alignment files were binned using multiBamSummary function with default setting and Pearson correlation was calculated using plotCorrelation function. To confirm the quality of our ChIP-seq data, we also compared our ChIP-seq H3K4me3 and H3K27me3 using E14 NSC with the published ChIP-seq data H3K4me3 (ENCSR172XOZ) and H3K27me3 (ENCSR831YAX) using E14 forebrain (https://www.encodeproject.org/).

ChromHMM (1.14)[74] was used to evaluate the state transition between different stages. The alignment files of H3K4me3 and H3K27me3 in each stage were binned into 200-bp bins using BinarizeBam. Then, we established the model with 4 emission states (H3K4me3-only, H3K27me3-only, bivalent, and none) and trained with binned data using LearnModel command. These segmentation files with state information were used to plot alluvial plotting.

DiffBind package (2.10.0)[75] in R was used to find peaks with different intensities between samples and different stages and to visualize data with PCA. To do so, overlapping peaks among each sample were isolated and sequenced reads on these consensus peaks were counted using the dba.count function. As a result, a matrix in which each column indicates a consensus peak and each row indicates the normalized reads count was produced. The matrix was used to plot PCA using the ggplot function in R. The peaks with different intensities were calculated using dba.contrast and dba.analyze function. The MA plot was drawn using the dba.plotMA function.

**Library preparation and sequencing.** Library preparation for RNA sequencing (RNA-seq) was performed as described[76] using the TruSeq Stranded mRNA Library Prep Kit (Illumina) and 100–110 ng of total RNA. Library preparation for ChIP-sequencing (ChIP-seq) was performed as previously described[77] using the KAPA LTP Library Preparation Kit (KAPA Biosystems) and 2 ng of input DNA, or the entire amount of the ChIP DNA obtained. The KAPA Real-Time Library Amplification Kit (KAPA Biosystems) was used in conjunction with the library preparation kits (TruSeq Stranded mRNA Library Prep Kit) to minimize the number of PCR cycles for library amplification. Library preparation for single-cell RNA-seq (scRNA-seq) was performed following the standard protocol of the 10x Genomics Chromium Single Cell 3′ v2 Kit (10x Genomics). Sequencing was performed in HiSeq1500 (Illumina) with the HiSeq SR Rapid Cluster Kit v2 (Illumina), the HiSeq PE Rapid Cluster Kit v2 (Illumina), or the TruSeq PE Cluster Kit v3-cBot-HS, to obtain single-end 80 nt reads for RNA-seq and ChIP-seq libraries, or paired-end 26 nt (Read 1)- 98 nt (Read 2) reads for scRNA-seq libraries. The total reads of each sample are listed in Supplementary Data 7.

**Polysome profiling.** Polysome profiling was performed as described previously[78] with the following modifications. After the knockdown of Fbl for 3 days from E10 as described in the section above, cells were lysed in 250 μl lysis buffer as described in the "Ribosome profiling" section. Two hundred μl of lysates were loaded on top of the 10–50% (w/v) sucrose gradient and centrifuged at 221,230 × g for 2.5 h at 4 °C using a rotor P40ST (Hitachi Koki). The sample was fractioned using Gradient Station (BioCamp) and the OD₂₅₄ spectrum was monitored with a Bio-mini UV monitor (ATTO). The fractions corresponding to the subpolysomes and polysomes were pooled and subjected to RNA extraction by Trizol LS (Thermo Fisher Scientific, 10296-010) and the Direct-zol RNA microprep kit (Zymo Research, R2060).

The RNAs were treated with 2 U of TURBO DNase for 30 min at 37 °C. The *Ezh2* mRNA was quantified with ReverTra Ace qPCR Master Mix (TOYOBO, FSQ-201) in a Thermal Cycler Dice Real-Time System Single (TaKaRa, TP850) with TB Green Premix Ex Taq II (Tli RNaseH Plus) (TaKaRa, RR820) and following primers: *Ezh2* forward: 5′-CCATGCAACACCCAACACATATAAG-3′; *Ezh2* reverse: 5′-GTAAGAGCAGCAGCAAACTCCTTA-3′.

**O-propargyl-puromycin (OPP) visualization.** Cells from E14 dorsal cortices with different genotypes were isolated and cultured for one day. Two microliters of OPP reagent was added to the medium and incubated for 30 min. These cells were fixed with 1% PFA overnight and OPP signals were detected using the Click-iT™ Plus OPP Alexa Fluor™ 488 Protein Synthesis Assay Kit (Invitrogen) as indicated by the manufacturer's protocol. To distinguish NSCs, immunostaining was performed with Sox2 antibody. The OPP signal was measured in Sox2 positive cells using CellProfiler (2.1.1)[79].

**Protein stability measurement.** Cells from E11 dorsal cortices were treated with 100 μg/ml cycloheximide for 0.5, 1, and 2 h followed by collection for western blot. As a control, cells without treatment were also collected.

**Knockout of 5′UTR of *Ezh2* and *Kdm6b*.** Two micrograms of pCAG-EGFP plasmid with 1 μg of gRNA plasmid for 5′UTR of *Ezh2* and *Kdm6b* and 1 μg of pCAX-Cas9 was transfected into cells from E11 cortices with NEON and these cells were cultured as mentioned above. Two days after transfection, the GFP-positive cells were sorted using the SH800 (SONY). These cells were used for reverse transcription (RT)-quantitative (Q) PCR, western blotting. and genotyping. For cells used for genotyping, 5 μL of protein kinase K was added and incubated for 1 h. After the treatment at 98 °C for 5 min, genotyping was performed used following primer: *Ezh2* forward: 5′-GAATTCTGCAGTCGACGCTTGATAGTGCTGGGGG G-3′ *Ezh2* reverse: 5′-CCGCGGTACCGTCGACGCCGAAGACTGGCCAGGC-3′ *Kdm6b* forward: 5′-GAATTCTGCAGTCGAGGCCTGGGTGCTGGATTTG-3′ *Kdm6b* reverse: 5′-CCGCGGTACCGTCGATCAGCATCAAGAGCCCCTAG-3′. The PCR products of these primers were cloned into a pCAG plasmid cut with SalI and sequencing. For the cells used for RT-qPCR, total RNA was extracted using RNeasy Mini Kits (Qiagen), and genomic DNA was removed by DNase treatment (Qiagen). RNA (30 ng) was used for the synthesis of cDNA with the PrimerScript RT reagent kit with gDNA eraser (TAKARA BIO). qPCR was performed with TB Green Premix Ex-taq II (TAKARA BIO). *Gapdh* was used as an internal reference to normalize the dosage of *Ehz2* and *Kdm6b* mRNA level, following primers were used for qPCR: *Ezh2* forward: 5′-GAGCGTATAAAGACACCACCTAAAC-3′; *Ezh2* reverse: 5′-CTCTGTCACTGTCTGTATCCTTTG-3′. *Kdm6b* forward: 5′-CC CCCATTTCAGCTGACTAA-3′; *Kdm6b* reverse: 5′-CTGGACCAAGGGGTGT GTT-3′; *Gapdh* forward: 5′-ATGAATACGGCTACAGCAACAGG-3′; *Gapdh* reverse: 5′-CTCTTGCTCAGTGTCCTTGCTG-3′.

The cells used for western blot were treated following the method in the "western blotting analysis" section.

**Motif find and calculation of minimum free energy.** To find the motif of mRNAs that downregulate or upregulate their translational efficiency after knockout of *Fbl*, 5′ UTR sequence of selected genes was initially extracted from Table Browser (UCSC) http://genome.ucsc.edu/cgi-bin/hgTables. Next, HOMER was used to extract enriched motif by findMotifsGenome.pl function with size = 200. For each 5′ UTR, minimum free energy was calculated every 50 bp by ViennaRNA Package 2.4.14[80] with RNALfold L50 −g, and the minimum value was used.

**Statistics and reproducibility.** Multiple comparisons for cell counting and 5′UTR knockout experiments were performed using one-way ANOVA followed by a Tukey's HSD. Student *t* test was used to test cap-independent translational activity after knockdown of *Fbl*. To test whether methylation level on rRNA was reduced in DKO comparing with control cells, a one-sided Wilcoxon signed-rank test was used. Kruskal–Wallis test with Dunn's multiple-comparison test was used to test different PC contributions of genes and global protein level among different genotypes of mice. The Wilcoxon signed-rank test was used to test different reads of H3K27me3 and H3K4me3 in selected genes and MEF of different gene groups. R, RStudio, and Excel (Microsoft) were used for these analyses. All of the experiments have been repeated at least twice.

**Reporting summary.** Further information on research design is available in the Nature Research Reporting Summary linked to this article.

## Data availability

The data that support this study are available from the corresponding authors upon reasonable request. All raw and proceeded data for scRNA, ChIP-seq and Ribo-seq generated in this study have been deposited in DNA Data Bank of Japan with accession number DRA009567, DRA009569 and DRA009729, DRA010791, DRA013203, and E-GEAD-348, E-GEAD-349, E-GEAD-350, and E-GEAD-467 in Bioproject PRJDB9278. Processed data generated in this study are also provided in the Supplementary Data files.

We compared our scRNA data with previous data GSE107122[18]. To confirm the quality of our ChIP-seq data, we also compared our ChIP-seq H3K4me3 and H3K27me3 using E14 NSC with the published ChIP-seq data H3K4me3 (ENCSR172XOZ) and H3K27me3 (ENCSR831YAX) using E14 forebrain. Source data are provided with this paper.

## Code availability

Code for calculation birthdate and differentiation score was deposited at https://github.com/wuquan723/Birthdate-and-differentiation-score and https://doi.org/10.5281/zenodo.5784971.

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

## Acknowledgements

We thank A. Shitamukai, I. Fujita, Y. Tsunekawa, D. Konno, and all members in Matsuzaki laboratory for their discussion and technologic advices. We thank C. Tanegashima, K. Tatsumi, and O. Nishimura of the Laboratory for Phyloinformatics, RIKEN Center for Biosystems Dynamics Research (BDR) for NGS library preparation, sequencing, and assistance on data production and analysis. This work was supported by the Japan Society for the Promotion of Science (JSPS) Grants-in-Aid for Scientific Research (KAKENHI) grand no. 18K14722 to Q.W., Scientific Research on Innovative Areas 17H05779 and 19H04791, and RIKEN funds to F.M. Q.W. was also supported by JSPS Postdoctoral Fellowship and RIKEN Special Postdoctoral Researcher Program.

## Author contributions

F.M. supervised the project. F.M. and Q.W. designed the experiments and wrote the manuscript. Q.W. carried out experiments and performed the bioinformatics analysis of

data. Y.S. and S.I. developed the methods for ribosome profiling using dorsal cortices and helped Q.W. in the analysis of ribosome profiling data. Y.S. and S.I. performed polysome profiling. T.A. and H.K. generated *Fbl* conditional knockout mice. T.S. did in utero electroporation. A.O. helped Q.W. in sequencing, qPCR, and purification of plasmids.

## Competing interests

The authors declare no competing interests.
