## [Peer Review File · Nature Communications]

REVIEWER COMMENTS

Reviewer #1 (Remarks to the Author):

The authors reported the discovery of fibrillarin (Fbl) as a regulator of cortical development. They compared single cell RNA-seq (scRNA-seq) profiles of neural progenitor cells (NPCs) at embryonic days 11 and 14 and identified Fbl as one of the genes that were significantly more highly expressed in NPCs at E11 than those at E14. Depletion of Fbl in the mouse dorsal cortex led to microcephaly and NPCs failing to differentiate to neurons or glia. Using the expression of 95 genes from scRNA-seq profiles, Fbl-depleted NPCs appeared to be developmentally delayed than control NPCs. These developmental phenotypes may be explained by the observed requirement of Fbl in G1 to S phase progression. ChIP-seq profiling of H3K27me3 and H3K4me3 in Hes1-positive NPCs at E11, E12, and E14 suggest dramatic H3K27me3 changes and modest H3K4me3 changes. Fbl affects H3K27me3 distribution. Mechanistic studies revealed that Fbl indirectly affects the translation of 2 H3K27 modifiers, Ezh2 methyltransferase and Kdm6b demethylase. This regulation is likely mediated through the 5' untranslated region (UTR) of Ezh2 and Kdm6a transcripts. Together, authors provided provocative data suggesting that Fbl affects cortical development through its regulation of Ezh2 and Kdm6a.

This is a data-rich transcript that requires extensive revision. Authors may consider rearranging the order of some data to improve messaging of the main points. Current data do not sufficiently support that Fbl affects the developmental clock of embryonic NPCs. Overall, authors may consider expand the description of rationale and experimental data to improve readability. Points listed below are mostly ordered along data presented in the manuscript.

1. Introduction needs some information about Kdm6b in brain development.
2. scRNA-seq data serve as the foundation of this manuscript. Authors may consider expanding the explanation of what's done throughout the manuscript but especially the scRNA-seq. Examples are how many cells were analyzed per developmental point, gender differences (since XIST is markedly differentially expressed), and ways to quality control the data. Related to the last point, authors used a part of Extended Data Fig 1 to show quality of ChIP-seq data but no explanation for scRNA-seq.
3. Extended Data Fig 1b-f are hard to see. Scales in Extended Data Fig 1f are not clear.
4. How do genes associated with H3K27me3 and H3K4me3 changes correlate to differentially expressed genes in E11 vs. E14 NPCs?
5. For all principal component analysis, only comparisons of PC1 and PC2 were presented, which account for 35% to 50% differences. For example, Fig. 1l PC1 vs. PC2 account for 37.7 (12.1+25.5) % differences.
6. For Fig. 1m, gene ontology of genes associated with H3K27me3 changes may provide functional explanation.
7. In line 137, the subheading does not reflect content of the section. Data did not support Fbl as a key regulator of temporal patterning. In line 138, please consider changing "asked what factors promote" to something similar to "search for factors associated with". For rest of the manuscript, there are other incidences of potential over-reaching.
8. Authors may consider transitioning data from Fig. 1a-c to Fig 2a. and put rest of Fig. 1 later.
9. It is quite concerning about the use of Trp53-KO in combination with Fbl for all data analyses throughout the manuscript. The inclusion of Trp53-KO is confounding analysis of Fbl's role. Although authors stated that Trp53 did not impact brain size or neurogenesis (Extended Data Fig 3a-b), published literature provides strong evidence that Trp53 affects neurogenesis and brain development.
10. In Fig. 2g-I, please define "bins" in y-axis. In Fig. 2k, please define "section in y-axis."
11. The dysfunction of temporal patterning factor is expected to disrupt temporal patterning and result in shifting of neurogenesis and gliogenesis phases. However, Fbl-Trp53 co-depletion resulted in failure of differentiation of neurons as well as glia. This kind of differentiation failure is not characteristic of temporal patterning defects. Fbl-Trp53 co-depletion likely renders NPCs unable to execute differentiation programs in general – this is also supported by the developmental delay of DKO NPCs.
12. In the section starting from line 186, there are concerns about data analysis. Extended data table

- 3a repeats gene names. Example repeats are Hmga2, Neurod2, and Neurod6.
13. Conclusion in lines 2110-2111 is contradicted by data from Fig 3f, which suggest that E14 DKO is closer to control in pseudotime.
 14. Extended Data Fig 4b appear to not support statement in lines 212-213 since it lists GO terms and not related to the mathematical modeling.
 15. There are spelling mistakes such as "NCSs" in line 224.
 16. Words, "directly" in line 263 and "selectively" in line 275 are better to be removed. The use of these 2 terms suggests that Fbl directly binds the mRNAs of Ezh2 and Kdm6b and selects them for translation. Data presented do not support that.
 17. Does "higher" in line 284 pass statistical test?
 18. About lines 286-287. (a) Authors may consider rephrasing "these genes were not upregulated" to "failed to upregulate" when comparing E14 DKO to control NPCs. Extended Fig. 4d did not support the implication (started from lines 283 to 287) that genes with increased H3K27me3 at TSSs failed to upregulate in DKO NPCs. Fig. 4d only listed top 40 genes that were differentially expressed.
 19. In Fig. 6f, the E14 and E14 \rightarrow _DKO dots were hard to distinguish. Authors may consider changing colors to be more mindful of colorblind readers.
 20. First sentence in lines 292-293 is confusing. Sentence in lines 294-297 is best to be split into 2-3 sentences to improve readability.
 21. Explanation of the inhibitor treatment needs more explanation. For example, which cells at what developmental stage were used? What dosages of the inhibitors were used? Citations for the inhibitors? How long was the treatment?
 22. Importantly, separate inhibition of Ezh2 or Kdm6a/b (GSK-J4 targets both and potentially other Kdms) resulted in relatively few genes affected, 210 and 409 (line 298). However, combining both inhibitors affected 10056 genes. Such a drastic disruption to gene expression suggests cells were severely compromised, potentially dying. Were the treatment so severely disrupted cell physiology that the calculated scores (birthdate and differentiation) were brought down; therefore, the effect of combining inhibitors is not specific to Ezh2 and Kdm6b,
 23. Author may consider rephrase "Fbl target mRNAs" to "mRNAs affected by Fbl depletion".
 24. It is not clear that the sentence starting in "Since H3K27me3..." in lines 354-357 is related to the message of this manuscript. This manuscript purports to describe the specific regulator, Fbl, in contrast to the message in lines 354-357.
 25. Model about rRNA modification by Fbl as a mechanism is not supported by the presented data. Is there inhibitor specific to the methyltransferase activity of Fbl?
 26. Ezh2 is under strong post-transcriptional regulation, including that by miRNAs. In discussion, authors may consider discussion of known post-translational regulation of Ezh2.

Reviewer #2 (Remarks to the Author):

Wu et al use epigenetic profiling of neural stem cell differentiation in the mouse to identify genes that temporally impact NSC patterning. They identify a module of genes related to the ribosome that change in expression between NSC developmental stages. From this module, they focus on the gene Fibrillarin (Fbl), a ribosomal rRNA methyltransferase. They then go on to show that Fbl regulates translation of chromatin modifiers and that its depletion affects the developmental clock. They suggest that Fbl affects translation through a cap-independent mechanism. Overall, the paper is a good contribution, but this reviewer felt that the final portion re: cap-independent translation was insufficiently supported, somewhat tangential, and unnecessary for publication of the remainder of the work subject to some additional clarifications as below.

Major concerns:

- 1) The justification for pursuing Fbl specifically should be explained better given that
 - a. It is ranked low amongst the module nodes in the brown module (Figure 2b) and

b. A large number of ribosomal protein genes (Figure 2a, Extended Data table 2 Rpl14, Rpl17, Rpl36, Rpl38, Rps15, Rps17, Rps21, Rps27, Rps28, Rps3, Rps6) were also identified in this gene module. Do the authors expect that the translational impact on NSC differentiation is specific to Fbl or a more general perturbation to translation/ribosome levels? If the authors wish to suggest that Fbl is specifically required for this, controls perturbing other ribosomal proteins are essential. If the authors are using Fbl as an example of genes in this module, the paper should be reworded to reflect that Fbl is a perturbation to translation as opposed to the pathway requiring Fbl specifically.

2) I was unable to find a methods section for how the flow cytometry experiments were performed and quantified.

3) As I understand it, the quantification in figure 7D is the percent of double positive cells in flow, as opposed to a quantitative measurement of fluorescence intensity. This may create a dependence on the location of the gate. For example, if the GFP gate was shifted slightly higher in Figure Extended 7a then it would substantially change the number of positive cells in the Fbl siRNA condition. This means that the gate location strongly determines the fold change between Fbl and control siRNAs in this analysis. Instead, plotting the ratio between GFP and BFP per cell could be used to set a threshold using e.g. gaussian decomposition in this ratio or in the raw FACS plots to more robustly define the population boundaries.

4) In Figure S7D, BFP fluorescence of the Ezh2 reporter seems to increase upon Fbl depletion? What does the GFP signal look like? Is the decrease in the ratio based on the increase in the denominator, namely the BFP signal?

5) Broadly, the beginning of this paper is very strong and implicates translation and perhaps Fbl specifically in NSC differentiation. The final experiments that try to claim cap-independent translation is involved fall short. The stated reason for invoking cap-independent translation is the presence of poly U motifs and RNA structure, but there are many mRNAs with these properties that may not drive cap-independent translation. Mills & Green (2017, Science) discussed how general perturbations to translation can selectively impact mRNAs with inhibitory elements such as mRNA structure, so it is not clear that a cap-independent mechanism is needed to explain these findings. The bicistronic reporter used to test this model is problematic and the results could equally be explained by differential stop codon readthrough or other mechanisms. This reviewer feels that ending with a description of the properties of the translationally impacted genes is appropriate and welcome but that substantially more work is required to show it is cap-independent, that the current experiments are too superficial, and that the paper would be stronger without trying to prove this specific mechanism – which seems beyond the scope of this paper.

Minor concerns:

What do the flow plots for the "positive control" IRES (Cdkn1b) look like? If the authors are claiming that 1-3% GFP signal is sufficient to presume IRES usage, then it is important to show what the %GFP positive cells in their positive control is, and if possible, how it compares to a bona fide viral IRES to get an idea of dynamic range.

Reviewer #3 (Remarks to the Author):

The authors have characterized the temporal gene regulation during neural stem cells progression. The authors focused in the H3K27me3 modifiers, and in particular Fbl. Fbl looks to regulated mRNA translation, likely by recognizing particular sequences in the 5'UTR. Using several state-of-the-art

techniques, (e.g. single cell sequencing, ribo-seq, Chip-seq), this work emerges as an interesting example of gene regulation at multiple level, translation, transcription and mRNA level.

I would like to mention some comments that were unclear and/or might improve the work.

-Figure 1. Page 5, Lines 110 to 120. It is unclear to me, what is the relation between the changes in H3K27me3 and H3K4me3 and RNAlevel between E11 to E14. Specially thinking in Fig 1d and how the author selected the 1505 and 20 sites.

-Page 7, line 153. The western blot suggesting that Fbl protein is higher in E11 than in E14, is the first experimental data suggesting that translation regulation might be playing a role. Thus, I would show the Western blot as main figure. While I was able to find the blot and the quantification (Extended Data Fig 2c), and one of the replicates does not look great. I was not able to find (Extended Data Fig 2d).

-Across the text, there is almost no statistical information. Just to mention one example, in page 7, line 158 it says "dramatic brain size reduction", I am not a mouse expert and do not see a "dramatic brain size reduction". Please quantify it and add a P value. At least in the files that I downloaded; I found the Extended Data Fig 2f but I could not find Extended Data Fig 2g.

-Page 11, line 255. The authors found a decreased level of synthesized protein in the DKO NSC (Extended Data fig 6a). Honestly, it is very hard to find the data, there are Extended Data Table6a and Extended _Fig6a. I am not sure which one I need to see, but in any case, I would expect to find a plot (barplot?) with a P value calculated and not tables. Moreover, I think this is also a very important result and therefore, I would show it as main figure.

Reviewer #4 (Remarks to the Author):

The authors show that H3K4me3 and H3K27me3 modification changes during temporal patterning of NSCs. They show that depletion of Fbl (fibrillarin), a rRNA 2'-O-methyltransferase, results in impaired translation of both the Ezh2 methyltransferase and Kdm6b demethylase of H3K27me3. This engenders a delay in progression of the NSC state, thereby impeding brain development. The authors argue that Fbl selectively enhances the translation of H3K27me3 modifiers via a cap-independent mechanism. Whilst the authors present a substantial body of work, there is much which is missing regarding the mechanism of Fbl-mediated translational control. The discussion is highly speculative, as there is no mention of supportive data in the literature. There are also several issues regarding the text and other concerns outlined below.

Major concerns:

1. The authors claim that Fbl regulates translation of genes involved in H3K27me3 modification in a cap-independent manner (in the heading of the section it is stated "cap-dependent"). The evidence provided by the authors for a cap-independent mechanism is not convincing. First, the data in Fig. 7D should not be presented as a ratio of GFP to BFP signal as one could readily conclude that a lower value in the Fbl siRNA KO condition is simply due to more BFP signal (i.e. cap-dependent translation). This indeed looks like the case for Ezh2 (Extended Fig. 7D). Both GFP and BFP values should be graphed together. Second, there is no dedicated section in the methods detailing how the constructs for each gene were made and how these experiments were performed making it difficult for the reader to interpret the data. Lastly, to convincingly demonstrate that Fbl deletion reduces the translation of

Ezh2 and Kdm6b, the authors should perform polysome profiling to show a shift in these mRNAs towards lighter polysome fractions.

They characterize Ezh2 and Kdm6b IRESs only by using bicistronic plasmids, which is prone to misinterpretation [Jackson, R.J. (2013). The current status of vertebrate cellular mRNA IRESs. Cold Spring Harb. Perspect. Biol. 5, pii: a011569]. One common artefact with this approach is the generation of monocistronic mRNAs due to the presence cryptic promoters. Their RT-PCR analysis using primers in BFP and GFP regions is misleading, as it cannot detect such mRNAs (Extended Data, Fig. 7c). Northern blot with a GFP probe or RT-PCR using primers within GFP coding region should be provided. The most rigorous approach would be targeting the BFP cistron coding sequence with siRNA. If this siRNA does not reduce the expression of GFP cistron, then this expression is most likely driven by monocistronic mRNA (Van Eden et al. 2004.RNA 10:720-730). Alternatively, they can generate bicistronic mRNAs and monitor the translation of both cistrons after transfection of these mRNAs into cells. Also, the ability poly(U) to confer upon 5'UTRs an IRES activity is not well studied.

2. The discussion of the mechanistic aspect of the paper vis-à-vis translation is severely following speculative. For example, "In addition, it is likely that Fbl affects translation via the structure of 5'UTR of target mRNAs, which restricts the range of translational regulation, eventually generating the specificity of Fbl targets." It is unclear how an rRNA processing enzyme could affect the structure of the 5'UTR of target mRNAs to promote translation. The authors also have not shown how methylation of rRNA by Fbl confers specificity for the translational regulation of specific mRNAs, as depicted in Fig. 8. The notion that a marginal increase of methylation on some rRNA sites by Fbl (Extended Data Fig. 3d) changes the translational specificity of ribosomes in favour of Ezh2 and Kdm6b IRESs is extremely tenuous, as no experimental support and even mechanistic explanation is provided. Moreover, it is very baffling that the translation from Cdkn 1b, Ezh2 and Kdm6b IRESs is reduced after Fbl knockdown (Figure 7d). In fact, Fbl depletion is supposed to provide a competitive advantage over cellular IRESs, as it inhibits global translation, which is mainly cap-dependent (Extended Data Fig. 6a).

Other concerns:

1. Quantification of the brain images (Fig. 2e) should be included to confirm microcephaly.
2. The choice for using EMX1-Cre heterozygous instead of homozygous mice is not clear.
3. Fig. 3D, clarification is needed as to what the arrows are representing.
4. Fig. 3 E and F, components 1 and 2 need to be described.
5. Fig. 4E, the values indicated as percent in the graph are illegible.
6. The western blot in Fig. 5D is of very poor quality. The columns should be in line and each composite membrane zoomed to the same magnification. These issues apply to other westerns in the manuscript.
7. The authors should include representative images for the O-propargyl-puromycin experiment.
8. The exons flanked by loxP sites need to be mentioned.
9. The dilution for each antibody used should be included in the methods as well as the final concentration of protease inhibitor used for western blot.
10. The authors should provide a brief description as to how the 10X-Genomics single cell seq works and provide a reference if it has been successfully used in previous studies.
11. The manuscript requires thorough editing as there are many typos and grammatical errors.
12. Line spacing should be the same throughout the main text.

Summary of revised points:

We would thank all reviewers for the careful reading of our manuscript and for their constructive suggestions. Because we have revised our manuscript a great deal, we will first list the major changes prior to providing our ‘point-by-point’ responses, in order to help the reviewers better understand what we have changed. We have also marked all of changes in the manuscript.

Major revised points that we have made:

- 1) Reviewer 1 suggested that the *Fbl*/P53 double knockout mice that we used to prevent apoptosis might not be suitable for analysing the function of *Fbl*, because P53 may affect brain development. To avoid such a possible effect of the loss of P53 on brain development, we knocked down *Fbl* in neural stem cells (NSCs) in the wild type cells, and performed RNA-seq to confirm the function of *Fbl* was retained. Unlike *Fbl* conditional knockout mice, we found that knockdown of *Fbl* did not induce global apoptosis, and yet confirmed delayed temporal fate transitions as observed in double mutant mice. Thus, we concluded that the delayed temporal progression of NSCs was caused by the loss of *Fbl* with no influence on the P53 mutant background.
- 2) Reviewer 1 was also concerned about the specificity of chemical inhibitors against demethylase and methyltransferase, which we used in our study. This concern is based on the fact that “combining both inhibitors affected 10056 genes” and the reviewer suggested that such “a drastic disruption to gene expression indicates that cells were severely compromised, potentially dying”. We agree with the Reviewer’s concern about the specificity of the chemical inhibitors, and thus performed the same experiment under the condition, the dose of the Kdm6b inhibitor was reduced from 2.5 μM to 0.5 μM . Under these new conditions of Kdm6b 0.5 μM treatment, combining both inhibitors also led to the reduction of the birthdate score of NSCs, and only 2344 genes were affected. To investigate the specificity of each inhibitor, we extracted differentially expressed genes (DEGs) following inhibitor treatment and subjected the treated samples to Chip Enrichment analysis (Lachmann et al., 2010). This allowed us to compare the DEGs from this experiment with publicly available ChIP data to determine which factors exerted similar effects as the inhibitors. If the inhibitors were not specific and affected other pathways, we expected to detect these factors using this method. The results showed that in all cases (after GSK-J4 or GSK343 treatment, or after double inhibition), Suz12, a subunit of the PRC2 complex responsible for H3K27me3, was the only factor that showed similar changes in target genes as DEGs after inhibitors (GSK-J4 or GSK343 or both) were administered. These data suggested that our inhibitors specifically affected the modification of H3K27me3 at appropriate concentrations. We replaced the data for inhibitor experiments using those obtained at the lower concentration of GSK-J4.
- 3) As suggested by Reviewer 4, we performed polysome profiling to confirm the *Fbl* effects on translation of *Ezh2* mRNA and *Kdm6b* mRNA. Polysome profiling requires large numbers of cells (at least 10^7), but it is difficult to collect a sufficient amount of cells from embryonic brains. Therefore, we pooled fractions corresponding to subpolysomes and polysomes, after samples were fractionated. The results showed that when *Fbl* expression was knocked down, more *Ezh2* mRNA was observed in the subpolysome fraction than that in the polysome fraction, indicating that *Ezh2* translation was affected by the reduction of *Fbl*. Conversely, since the expression of *Kdm6b* was relatively low in NSCs, we could not identify their mRNAs in polysome profiling.
- 4) Reviewer 4 suggested that “the generation of monocistronic mRNAs due to the presence cryptic promoters” might give rise to an apparent polycistronic translation in the

experiment shown in Figure 7c in the previous manuscript, and that “The most rigorous approach to test this would be targeting the BFP cistron coding sequence with siRNA”. According to this suggestion, we used two different siRNAs to knockdown BFP. If *Gfp* mRNA is generated by cryptic promoters independent of *Bfp* mRNA (Figure 1B), knockdown of *Bfp* would not affect *Gfp* mRNA. In both cases, knockdown of *Bfp* mRNA led to a decrease in both *Bfp* and *Gfp* mRNA to a similar level (C in the Figure below). Thus, a *Gfp* mRNA is not generated by cryptic promoters independently of *Bfp* mRNA transcription. Therefore, the coding sequence for *Gfp* and that of *Bfp* are most likely translated from polycistronic mRNAs. Thus, if we follow the logic of the reviewer 4, the observed GFP signal is likely due to translation via 5'UTR of *Ezh2* in front of the *Gfp* coding sequence, although this fraction is less than 3% compared with the BFP signal. As we have already deleted Figure 7, we show the new data below.

Figure 1. We transfected E11 NSCs with a bicistronic reporter (BFP-*Ezh2*-5'UTR-GFP) together with control or BFP siRNA (two different types). After 2 days, we retrieved cells and performed qPCR to investigate the expression of BFP and GFP mRNA. Two different BFP siRNAs present different efficiencies in reduce BFP mRNA and they also reduced GFP mRNA to the same level as BFP mRNA (compare bars with same colours in left and right figures).

- 5) Reviewer 2 strongly suggested that deleting the statement suggesting the cap-independent mechanism would be a better option. We carefully discussed the reviewer’s suggestion with the coauthors, and we agreed with the reviewer to withdraw the claim, whereas we retained our claim that the 5'UTR of mRNAs is important for their translation in the revised manuscript. We provide four reasons why we felt it would be better to withdraw the statements regarding the cap-independent mechanism in the translational control by Fbl;
 - a) As shown in the original manuscript, only a small fraction of cells (~3%) could be translated in a cap-independent manner (previous Extended Figure 7a). As suggested

by Reviewer 2, we tested a bona fide viral IRES as the control. The viral IRES strongly induced GFP and almost all of BFP cells were GFP-positive (Following Figure). Therefore, the same mechanism did not fully explain how Fbl enhances the translation of Ezh2 and Kdm6b.

Figure 2 We transfected E11 NSCs with a bicistronic reporter (BFP-virus IRES-GFP). After 2 days, we observed that most of BFP positive cells (94.5%) were also GFP-positive, indicating the high activity of virus IRES.

b) We also discussed with the researchers who study Fbl function in human cells. They also observed a similar situation; the efficiency of Cap-independent translation of mRNAs is also lower than 10% (Frédéric CATEZ; personal communication, also see <https://doi.org/10.1073/pnas.1707674114>).

c) It is still unclear how rRNA methylation levels affect the 5'-UTR structure of target mRNAs to facilitate translation.

Therefore, we considered that whether the cap-independent mechanism interferes with Fbl function in brain development would be beyond the scope of the present study and warrants further investigation as suggested by Reviewer 2. We have incorporated these arguments into “Discussion”, and further compared other studies dealing with cap-independent mechanisms.

6) Finally, we added several control experiments and compared our experimental data with previous data to demonstrate the high quality of single cell data.

‘Point-by-point’ responses:

Reviewer #1 (Remarks to the Author):

The authors reported the discovery of fibrillarin (Fbl) as a regulator of cortical development. They compared single cell RNA-seq (scRNA-seq) profiles of neural progenitor cells (NPCs) at embryonic days 11 and 14 and identified Fbl as one of the genes that were significantly more highly expressed in NPCs at E11 than those at E14. Depletion of Fbl in the mouse dorsal cortex led to microcephaly and NPCs failing to differentiate to neurons or glia. Using the expression of 95 genes from scRNA-seq profiles, Fbl-depleted NPCs appeared to be developmentally delayed than control NPCs. These developmental phenotypes may be explained by the observed requirement of Fbl in G1 to S phase progression. ChIP-seq profiling of H3K27me3 and H3K4me3 in Hes1-positive NPCs at E11, E12, and E14 suggest dramatic H3K27me3 changes and modest H3K4me3 changes. Fbl affects H3K27me3 distribution. Mechanistic studies revealed that Fbl indirectly affects the translation of 2 H3K27 modifiers, Ezh2 methyltransferase and Kdm6b demethylase. This regulation is likely mediated through the 5' untranslated region (UTR) of Ezh2 and Kdm6a transcripts. Together, authors provided provocative data suggesting that Fbl affects cortical development through its regulation of Ezh2 and Kdm6a.

This is a data-rich transcript that requires extensive revision. Authors may consider rearranging the order of some data to improve messaging of the main points. Current data do not sufficiently support that Fbl affects the developmental clock of embryonic NPCs. Overall, authors may consider expand the description of rationale and experimental data to improve readability. Points listed below are mostly ordered along data presented in the manuscript. **Response to Reviewer.** We would thank the reviewer for the careful reading and constructive suggestions. As you will see below, we have revised the manuscript by adding new experiments and change the order and description of data, so that the rationale of the experiments and their results are more clearly understood and convincing.

1. Introduction needs some information about Kdm6b in brain development.

Response to Reviewer. We thank the reviewer for the suggestion. We have added the information about Kdm6b in brain development in the introduction as follows:

In the embryonic forebrain, a demethylase of H3K27me3, Kdm6b, is required for NSCs to activate neurogenic genes in response to retinoic acid (Jepsen et al., 2007). In embryonic stem cells, Kdm6b is essential for neural commitment (Burgold et al., 2008), whereas its function in temporal fate changes of NSCs has not been clarified. (Line 77-81)

2. scRNA-seq data serve as the foundation of this manuscript. Authors may consider expanding the explanation of what's done throughout the manuscript but especially the scRNA-seq. Examples are how many cells were analyzed per developmental point, gender differences (since XIST is markedly differentially expressed), and ways to quality control the data. Related to the last point, authors used a part of Extended Data Fig 1 to show quality of ChIP-seq data but no explanation for scRNA-seq.

Response to Reviewer. We would thank the reviewer for this suggestion. We have added the method for quality control in the Supplementary Figure1 and have added a detailed introduction in the manuscript. We also compared our data with a previous published scRNA-seq dataset from embryonic brains (Yuzwa et al., 2017). These descriptions were added into the section of “Analysis of brain development at single cell level”.

Please note that in our experiments, several embryo samples were pooled together, therefore, the differential expression of XIST does not include gender differences. Higher expression of Xist in NSCs at early stages than in later staged NSCs were also observed in other datasets (Telley et al., Science 364, eaav2522 (2019). Their data can be easily verified here: genebrowser.unige.ch/telagirdon/

3. Extended Data Fig 1b-f are hard to see. Scales in Extended Data Fig 1f are not clear.

Response to Reviewer. We apologize for the poor quality of the visual representation of the data. We have changed the scales showing the correlation to make it clear. In the revised manuscript, these data are shown in Supplementary Figure 8a-f, since we have followed Reviewer 1’s suggestion and changed the order of the figures (Point 8).

4. How do genes associated with H3K27me3 and H3K4me3 changes correlate to differentially expressed genes in E11 vs. E14 NPCs?

Response to Reviewer. We have added these data to Fig. 6k and 6l (also shown below in Figure R1-1). We extracted the H3K27me3-peaks which are specifically detected in E11 and E14 NPCs. Next, we assigned these peaks to specific genes using an online tool: GREAT (<http://great.stanford.edu/public/html/>). As a result, we identified 540 and 54 genes whose H3K27me3 peaks were specifically observed in E11 and E14 NPCs, respectively. We added genes names as Supplementary Table 5a,b. As the reviewer mentioned in point 6, we added GO terms for H3K27me3.

For H3K4me3, as only few changes could be detected (20 peaks change), we provide the gene names in Supplementary Table 5e, but did not present the associated GO terms.

Figure R1-1 Go term analysis of genes sharing genomic regions E11 (a) and E14 (b)-specific peaks of H3K27me3.

The interpretation of the results is described in the Section of “GO term analysis of genes associated with H3K27me3 changes” as follows;

“We observed many genes required for neuron differentiation and development are highly modified by H3K27me3 at E11 (Fig. 6k and Supplementary Table 5c). Repression of these genes by H3K27me3 are consistent with the stage E11 when NSCs proliferate with suppression of neuronal differentiation. At E14, in contrast to E11, we observed genes that regulated of proliferation of NSCs were modified by H3K27me3, consistent with the

neurogenic property of NSCs at this stage (Fig. 6l and Supplementary Table 5d). Line349-354

5. For all principal component analysis, only comparisons of PC1 and PC2 were presented, which account for 35% to 50% differences. For example, Fig. 11 PC1 vs. PC2 account for 37.7 (12.1+25.5) % differences.

Response to Reviewer. We apologize for our unclear explanation. In the revised manuscript, histograms showing the contribution of PC components are shown (Supplementary Fig.8 g and h). As you will see in this histogram, the contributions of PC1, PC2, and PC3 are higher than other PCs. Therefore, we used a 3D representation of PC1, PC2, and PC3, which accounted for 46.7% and 55.2 % for H3K4me3 and H3K27me3, respectively, in new Fig. 6i and 6j. These new data support our interpretation, as we claimed in the previous version of the manuscript, that H3K27me3 correlates with temporal progression, while H3K4me3 showed no apparent correlation.

Figure R1-2 Principal component analysis (PCA) of H3K27me3 (a) and H3K4me3 (b) peaks, showing H3K27me3 samples can represent developmental time.

6. For Fig. 1m, gene ontology of genes associated with H3K27me3 changes may provide functional explanation.

Response to Reviewer. We thank the reviewer for a constructive advice. We have added the GO information associated with H3K27me3 changes as indicated in our response to point 4 above, which is indeed useful for understanding the role of H3K27me3. These data are shown as new Figure 6k and 6l, and are described in the “GO term analysis of genes associated with H3K27me3 changes” in the revised manuscript.

7. In line 137, the subheading does not reflect content of the section. Data did not support *Fbl* as a key regulator of temporal patterning. In line 138, please consider changing “asked what factors promote” to something similar to “search for factors associated with”. For rest of the manuscript, there are other incidences of potential over-reaching.

Response to Reviewer. We are grateful to the reviewer for the suggestions, and we agree with the reviewer’s concerns. We have changed the subheading from “as a key regulator” to “as a candidate regulator”. We also changed this and other incidences of potential over-reaching within our manuscript accordingly as listed below:

a) In the old manuscript line 227, we changed the subheading “*Fbl* affects temporal patterning through H3K27m3 modification” to “Deletion of *Fbl* affects H3K27me3 modification in NSCs”

b) We remove the discussion in line 364 “We hypothesized that rRNA modification by Fbl facilitates ribosomes to recognize or bind 5'UTR of target genes, thereby enhancing their translation (Fig. 8)”, because we did not have any direct evidence to show how modification on rRNA affected translation. We have also modified Figure 8.

8. Authors may consider transitioning data from Fig. 1a-c to Fig 2a. and put rest of Fig. 1 later.

Response to Reviewer. We thank the reviewer for the suggestion. We have rearranged our manuscript. We put ChIP-seq data after ribosome profiling analysis.

9. It is quite concerning about the use of Trp53-KO in combination with Fbl for all data analyses throughout the manuscript. The inclusion of Trp53-KO is confounding analysis of Fbl's role. Although authors stated that Trp53 did not impact brain size or neurogenesis (Extended Data Fig 3a-b), published literature provides strong evidence that Trp53 affects neurogenesis and brain development.

Response to Reviewer. We thank reviewer for the critical suggestion, and we also understand the reviewer's concern. To directly evaluate the role of Fbl in neural stem cells, we knocked down *Fbl* expression using siRNA. The siRNA treatment reduced *Fbl* mRNA to 30%. In this context, we did not observe upregulation of the apoptosis related gene, Trp53. We then performed RNA-seq analysis and found that knockdown of *Fbl* led to downregulation of the birthdate score (Figure R-1-3a). We also observed the persistent of deep layer makers (*Bcl11b* and *Tbr1*) and delay of expression of upper layer markers (*Pou3f2*, *Pou3f3*, *Satb2* and *Zbtb20*), suggesting a delay of fate transition after knockdown of *Fbl* (Figure R-1-3b-d).

Conversely, as far as we known, a subset of *Trp53* knockout mice showed exencephaly and anencephaly because of the failure of neural tube closure (Armstrong et al., 1995; Sah et al., 1995). We also observed such findings mice, however, we did not sample them for analysis. We combined these with normal brains, which did not show any detectable changes phenotype in neurogenesis.

Figure R-1-3: Knockdown of *Fbl* by siRNA affects temporal fate transition of NSCs. (a) *Fbl* siRNA reduced the expression of *Fbl* but did not induce the expression *Trp53*. (b, c) The expression level change of deep layer neuron markers (b) and upper layer neuron markers (c). (d) Birthdate score after the *Fbl* siRNA treatment for 3 days from E10.5.

10. In Fig. 2g-I, please define “bins” in y-axis. In Fig. 2k, please define “section in y-axis.

Response to Reviewer. We thank reviewer for careful reading our manuscript. We counted an area of 200 μm in length along with the apical membrane and defined these areas as bins. We have revised the manuscript and added the definition in the figure legend as follows. Fig. 2 has become Fig.1.

“For each section, an area with the length of 200 μm along with apical membrane was counted. Such region was defined as a bin.”

11. The dysfunction of temporal patterning factor is expected to disrupt temporal patterning and result in shifting of neurogenesis and gliogenesis phases. However, *Fbl*-*Trp53* co-depletion resulted in failure of differentiation of neurons as well as glia. This kind of differentiation failure is not characteristic of temporal patterning defects. *Fbl*-*Trp53* co-depletion likely renders NPCs unable to execute differentiation programs in general – this is also supported by the developmental delay of DKO NPCs.

Response to Reviewer. We agree that DKO NPCs unable to execute differentiation programs. As we described in the manuscript, we considered that *Fbl* presents dual functions in both the promotion of differentiation and temporal fate transitions. Importantly, the differentiation process is independent of temporal progression. It has been reported that overexpression of NICD is sufficient to repress the differentiation process, however, it not sufficient to interrupt the temporal fate progressions. Once the NICD is removed, NSCs

resume generating neurons with those ones not presenting temporary inhibition of neurogenesis by Notch activation (Ken-ichi Mizutani and Tetsuichiro Saito, 2005). Thus, differentiation failure should be considered separately to temporal patterning defects.

Conversely, as the reviewer stated, previous studies have indirectly investigated the final products (neuron or glia) of NPCs to confirm the temporal fate of NPCs. However, in our study, because the deletion of *Fbl* influenced the differentiation of NPCs, we could not observe the same phenotype as previous studies. Instead, we directly evaluate the temporal fate of NPCs by calculating the respective birthdate scores based on scRNA-seq.

We have discussed this important issue in the Discussion part: “**Function of Fbl in temporal fate transitions is independent of cell cycle and differentiation**”.

12. In the section starting from line 186, there are concerns about data analysis. Extended data table 3a repeats gene names. Example repeats are Hmga2, Neurod2, and Neurod6.

Response to Reviewer. We apologize to the reviewer for the lack of a clear description in the Extended data Table 3a. These genes were marker genes for each cluster. If clusters are similar to each other, it is possible that they share the same marker genes. For example, NPCs are divided into three clusters in our data, likely because of their different cell cycle states. In this context, they share the same NPC marker genes such as Sox2 and Pax6.

13. Conclusion in lines 2110-211 is contradicted by data from Fig 3f, which suggest that E14 DKO is closer to control in pseudotime.

Response to Reviewer. We apologize that statement may have confused the Reviewer. We have changed this part to “compare with E14 *Fbl*^{+/+} or E14 *Fbl*^{Δ/+} cells, E14 DKO cells were closer to E12 *Fbl*^{Δ/+}” in the revised manuscript. We also added the time axes in Figure 2h. E14 *Fbl*^{+/+} or E14 *Fbl*^{Δ/+} cells were located at the terminal of the time axes, indicating their later temporal fate. However, DKO cells were located in the middle of the axes, indicating they are still in the transition state from early to later fate. Therefore, we concluded that there was a delay in temporal progression in DKO cells. These data are shown more clearly in Figure 2g-i.

14. Extended Data Fig 4b appear to not support statement in lines 212-213 since it lists GO terms and not related to the mathematical modeling.

Response to Reviewer. We apologize for our error. We have revised our manuscript and to make the statement clearer. A description of mathematical modeling has been added. We have changed the manuscript as follows:

To further confirm our results, we introduced a simple mathematical model to estimate the developmental time and differentiation state of each NSC (see methods for detail). We defined the birthdate score and the differentiation score of each cell as a weighted linear combination of specific temporal-axis and differentiation-axis genes, respectively¹². For example, the birthdate score of a cell is a sum of the weight of a temporal gene multiplied by its normalized expression level for all temporal genes (96 genes). The weight of a temporal gene is decided by the stage-dependent expression level and specificity of such gene; a gene which is higher expressed in late NSCs than early NSCs in most of late NSCs has higher weight. (Lines 217-224)

15. There are spelling mistakes such as “NCSs” in line 224.

Response to Reviewer. We apologize for the spelling errors and have corrected them.

16. Words, “directly” in line 263 and “selectively” in line 275 are better to be removed. The

use of these 2 terms suggests that Fbl directly binds the mRNAs of Ezh2 and Kdm6b and selects them for translation. Data presented do not support that.

Responding to Reviewer: We revised the text accordingly.

17. Does “higher” in line 284 pass statistical test?

Responding to Reviewer: We have added the result of statistical analysis using these data, which are now shown in Fig 7d and 7e, the abundance of H3K27me3 peaks in early-onset genes was significantly higher in the DKO samples, indicating the expression of these genes was repressed by H3K27me3

18. About lines 286-287. (a) Authors may consider rephrasing “these genes were not upregulated” to “failed to upregulate” when comparing E14 DKO to control NPCs. Extended Fig. 4d did not support the implication (started from lines 283 to 287) that genes with increased H3K27me3 at TSSs failed to upregulate in DKO NPCs. Fig. 4d only listed top 40 genes that were differentially expressed.

Response to Reviewer. We have revised the manuscript as the reviewer suggested. The data that show how the gene expression changed with increased H3K27me3 at TSSs in DKO NPCs in Supplementary Table 6e. In total, 137 genes were identified with increased H3K27me3 at TSSs in DKO NPCs, and of these 49 showed significantly lower expression in E14 DKO than in control NPCs. Only 6 genes showed significant higher expression in E14 DKO than control NPCs. We also the following description in the manuscript:
Indeed, the expression of most of these genes (49/137) failed to upregulate in NSCs of E14 DKO (Supplementary Table 6e).

19. In Fig. 6f, the E14 and E14⁻_DKO dots were hard to distinguish. Authors may consider changing colors to be more mindful of colorblind readers.

Response to Reviewer. We thank for reviewer’s suggestion. We have changed the shapes in the figure (revised in Fig. 7f), instead of changing the colour of the points.

20. First sentence in lines 292-293 is confusing. Sentence in lines 294-297 is best to be split into 2-3 sentences to improve readability.

Response to Reviewer. We have split the sentence into 2 sentences, accordingly.

Previous manuscript:

“To this end, we inhibited both methyltransferase and demethylase using the specific inhibitors GSK-343 and GSK-J4, respectively, investigated gene expression changes involved in birthdate and differentiation, and calculated the birthdate and differentiation scores after RNA-seq.

Current manuscript:

To confirm the role of H3K27me3 modification in temporal patterning, we collected cells from the E10 cortex and treated these cells with an inhibitor of methyltransferase: GSK-343 (2.5 μM) or a potential inhibitor of demethylase: GSK-J4 (0.5 μM) or both of these inhibitors for 3 days^{37,38}. We then investigated gene expression changes after treatment with inhibitors by RNA-seq.

21. Explanation of the inhibitor treatment needs more explanation. For example, which cells at what developmental stage were used? What dosages of the inhibitors were used? Citations for the inhibitors? How long was the treatment?

Response to Reviewer. As we also discuss in the following point, we repeated these experiments and describe them more clearly. We collected cells from the cortex at E10 and treated them with 2.5 μ M Gsk_343 and 0.5 μ M Gsk_j4 for three days. We also added citations for the inhibitors in ref.37 and ref.38.

22. Importantly, separate inhibition of Ezh2 or Kdm6a/b (GSK-J4 targets both and potentially other Kdms) resulted in relatively few genes affected, 210 and 409 (line 298). However, combining both inhibitors affected 10056 genes. Such a drastic disruption to gene expression suggests cells were severely compromised, potentially dying. Were the treatment so severely disrupted cell physiology that the calculated scores (birthdate and differentiation) were brought down; therefore, the effect of combining inhibitors is not specific to Ezh2 and Kdm6b,

Response to Reviewer. We understand the reviewer's concern. We reduced the dose of demethylase inhibitor from 2.5 μ M to 0.5 μ M and repeated the experiment (Inhibitor for methyltransferase: Gsk_343 was not changed from 2.5 μ M). We cultured cells from E10 for three days. In the new conditions, the combination of both inhibitors also led to a reduction of the birthdate score of NSCs, while only 2344 genes were affected. To investigate the specificity of each inhibitor, we extracted the differentially expressed genes (DEGs) after treatment with inhibitors and subjected the samples to Chip Enrichment analysis (Lachmann et al., 2010). This allowed us to compare the DEGs from this experiment with publicly available ChIP data to determine what factors had similar target genes as the DEGs. The results showed that in all cases (after GSKJ4 or GSK343 treatment, or after double inhibition), Suz12, a subunit of the PRC2 complex responsible for H3K27me3, was the only factor that share similar target genes as the DEGs. These data suggest that our specifically of the inhibitors influenced the modification of H3K27me3 at the concentrations used.

Figure R1-4 Inhibition of H3K27me3 methyltransferase and demethylase. Genes, which showed expression changes after exposure to methyltransferase inhibitor: GSK_343 (a), demethylase: GSK_J4 (b) or both (c), were also found to be the target of Suz12. (d) Inhibition of methyltransferase and demethylase exerted additive effects on temporal fate transition.

23. Author may consider rephrase “Fbl target mRNAs” to “mRNAs affected by Fbl depletion”.

Response to Reviewer. We have revised the phrase accordingly.

24. It is not clear that the sentence starting in “Since H3K27me3...” in lines 354-357 is related to the message of this manuscript. This manuscript purports to describe the specific regulator, Fbl, in contrast to the message in lines 354-357.

Response to Reviewer. We have deleted the sentence .

25. Model about rRNA modification by Fbl as a mechanism is not supported by the presented data. Is there inhibitor specific to the methyltransferase activity of Fbl?

Response to Reviewer. Unfortunately, to date there is no specific inhibitor for the methyltransferase activity of Fbl. We agree with that we do not have direct data to support the idea that rRNA modification by Fbl regulates translation. Fbl binds with the Ezh2, mRNA methyltransferase, we have no evidence that how Fbl controls the translation of Ezh2. There are several possibilities: rRNA, mRNA, and direct binding to Ezh2. We have added these mechanisms to the Discussion part: Possible mechanisms for Fbl to regulate translation.

As suggested by another reviewer who commented that “the paper would be stronger without trying to prove this specific mechanism – which seems beyond the scope of this paper”, we

have deleted this last part showing Fbl works as cap-independent mechanism and have discussed the mechanism more carefully.

26. Ezh2 is under strong post-transcriptional regulation, including that by miRNAs. In discussion, authors may consider discussion of known post-translational regulation of Ezh2.

Response to Reviewer. We thank for reviewer's suggestion and we added the description of post-transcription regulation of Ezh2.

"The post-transcription regulation of Ezh2 by microRNA is also involved in the differentiation of adult NSCs in the dentate gyrus of the hippocampus, the maturation of the neuron in the hippocampus, and the fate choice between neuronal and astrocyte differentiation in embryonic NSCs."

References:

Jepsen, K., Solum, D., Zhou, T., McEvelly, R.J., Kim, H.-J., Glass, C.K., Hermanson, O., and Rosenfeld, M.G. (2007). SMRT-mediated repression of an H3K27 demethylase in progression from neural stem cell to neuron. *Nature* 450, 415–419.

Burgold, T., Spreafico, F., De Santa, F., Totaro, M.G., Prosperini, E., Natoli, G., and Testa, G. (2008). The histone H3 lysine 27-specific demethylase Jmjd3 is required for neural commitment. *PLoS ONE* 3, e3034.

Telley, L., Agirman, G., Prados, J., Amberg, N., Fièvre, S., Oberst, P., Bartolini, G., Vitali, I., Cadilhac, C., Hippenmeyer, S., et al. (2019). Temporal patterning of apical progenitors and their daughter neurons in the developing neocortex. *Science* 364, eaav2522.

Yuzwa, S.A., Borrett, M.J., Innes, B.T., Voronova, A., Ketela, T., Kaplan, D.R., Bader, G.D., and Miller, F.D. (2017). Developmental Emergence of Adult Neural Stem Cells as Revealed by Single-Cell Transcriptional Profiling. *CellReports* 21, 3970–3986.

Sah VP, Attardi LD, Mulligan GJ, Williams BO, Bronson RT, Jacks T. A subset of p53-deficient embryos exhibit exencephaly. *Nat Genet.* 1995 Jun;10(2):175-80. doi: 10.1038/ng0695-175. PMID: 7663512.

Armstrong, J.F., Kaufman, M.H., Harrison, D.J., and Clarke, A.R. (1995). High-frequency developmental abnormalities in p53-deficient mice. *Current Biology* 5, 931–936.

Mizutani, K.-I., and Saito, T. (2005). Progenitors resume generating neurons after temporary inhibition of neurogenesis by Notch activation in the mammalian cerebral cortex. *Development* 132, 1295–1304.

Kuleshov, M.V., Jones, M.R., Rouillard, A.D., Fernandez, N.F., Duan, Q., Wang, Z., Koplev, S., Jenkins, S.L., Jagodnik, K.M., Lachmann, A., et al. (2016). Enrichr: a comprehensive gene set enrichment analysis web server 2016 update. *Nucleic Acids Research* 44, W90–W97.

Reviewer #2 (Remarks to the Author):

Wu et al use epigenetic profiling of neural stem cell differentiation in the mouse to identify genes that temporally impact NSC patterning. They identify a module of genes related to the ribosome that change in expression between NSC developmental stages. From this module, they focus on the gene Fibrillarin (Fbl), a ribosomal rRNA methyltransferase. They then go on to show that Fbl regulates translation of chromatin modifiers and that its depletion affects the developmental clock. They suggest that Fbl affects translation through a cap-independent mechanism. Overall, the paper is a good contribution, but this reviewer felt that the final portion re: cap-independent translation was insufficiently supported, somewhat tangential, and unnecessary for publication of the remainder of the work subject to some additional clarifications as below.

Response to Reviewer. We would thank the reviewer for these critical suggestions and constructive criticism. We have discussed these concerns with researchers who focus on IRES-dependent mechanism by Fbl in human cells, who indicated that cap-independent translational regulation is also very weak in their experience. We used viral IRES to test our reporter as the reviewer's suggestion, we observed that most BFP-positive cell were also GFP positive (see below). As the reviewer indicated, the current data may not fully explain the dramatic decrease of Ezh2 and Kdm6b protein levels in DKO. Therefore, we agree with the reviewer and have remove the relevant part from the revised manuscript.

Major concerns:

1) The justification for pursuing Fbl specifically should be explained better given that
a. It is ranked low amongst the module nodes in the brown module (Figure 2b) and
b. A large number of ribosomal protein genes (Figure 2a, Extended Data table 2 Rpl14, Rpl17, Rpl36, Rpl38, Rps15, Rps17, Rps21, Rps27, Rps28, Rps3, Rps6) were also identified in this gene module. Do the authors expect that the translational impact on NSC differentiation is specific to Fbl or a more general perturbation to translation/ribosome levels? If the authors wish to suggest that Fbl is specifically required for this, controls perturbing other ribosomal proteins are essential. If the authors are using Fbl as an example of genes in this module, the paper should be reworded to reflect that Fbl is a perturbation to translation as opposed to the pathway requiring Fbl specifically.

Response to Reviewer. When we analyzed genes enriched in E11 NSCs, we found that many genes involved in translational regulation were enriched in E11 NSCs. Therefore, we considered that translational regulation was a general mechanism in early NSCs. Fbl is a component of this translational machinery. We revised the manuscript accordingly as follows: *“These results raise the possibility that translational regulation is important for temporal fate transitions of NSCs. To test this hypothesis, we particularly focus on Fbl (also known as Fibrillarin) as an example to investigate the impact of translational regulation during this process.”*

2) I was unable to find a methods section for how the flow cytometry experiments were performed and quantified.

Response to Reviewer. We apologize for lack of description. We have added a description in the method section: “Cell cycle analysis”.

3) As I understand it, the quantification in figure 7D is the percent of double positive cells in

flow, as opposed to a quantitative measurement of fluorescence intensity. This may create a dependence on the location of the gate. For example, if the GFP gate was shifted slightly higher in Figure Extended 7a then it would substantially change the number of positive cells in the Fbl siRNA condition. This means that the gate location strongly determines the fold change between Fbl and control siRNAs in this analysis. Instead, plotting the ratio between GFP and BFP per cell could be used to set a threshold using e.g. gaussian decomposition in this ratio or in the raw FACS plots to more robustly define the population boundaries.

Response to Reviewer. We agree with the reviewer that the gate location strongly determined the fold change between Fbl and control siRNA. It may be the best to determine the gate by gaussian decomposition of the signals. We deleted the experiments involving reporter assays from our manuscript as you strongly suggested (point no.5), we did not perform analysis as the reviewer's suggestion.

4) In Figure S7D, BFP fluorescence of the Ezh2 reporter seems to increase upon Fbl depletion? What does the GFP signal look like? Is the decrease in the ratio based on the increase in the denominator, namely the BFP signal?

Response to Reviewer. We could observe GFP signals using a microscope. The reason we used BMP to normalize GFP signal was that for every cell, the transfection efficiency is different. Cells with more vector (BMP signal) might have more possibility of expressing the GFP signal. We deleted the experiments relative to reporter assays from our manuscript as you strongly suggested (point no.5).

5) Broadly, the beginning of this paper is very strong and implicates translation and perhaps Fbl specifically in NSC differentiation. The final experiments that try to claim cap-independent translation is involved fall short. The stated reason for invoking cap-independent translation is the presence of poly U motifs and RNA structure, but there are many mRNAs with these properties that may not drive cap-independent translation. Mills & Green (2017, Science) discussed how general perturbations to translation can selectively impact mRNAs with inhibitory elements such as mRNA structure, so it is not clear that a cap-independent mechanism is needed to explain these findings. The bicistronic reporter used to test this model is problematic and the results could equally be explained by differential stop codon readthrough or other mechanisms. This reviewer feels that ending with a description of the properties of the translationally impacted genes is appropriate and welcome but that substantially more work is required to show it is cap-independent, that the current experiments are too superficial, and that the paper would be stronger without trying to prove this specific mechanism – which seems beyond the scope of this paper.

Response to Reviewer. As we indicated above, we agree with the reviewer and have removed the relative part from the revised manuscript. The problem regarding the experiment of bicistronic reporter was also pointed out by Reviewer 4. We performed experiments in response to Reviewer 4's concerns, and obtained reasonable results. Reviewer 4 suggested that "targeting the BFP cistron coding sequence with siRNA. If this siRNA does not reduce the expression of GFP cistron, then this expression is most likely driven by monocistronic mRNA." We also performed this experiment and found that BFP siRNA can reduce both BFP and GFP mRNA with the same efficiency, indicating the GFP signals do not derive from monocistronic mRNA. However, all authors extensively discussed whether we should include this experiment in the revised manuscript. We eventually reached the conclusion that this reporter assay was not sufficient to explain the *in vivo* situation, as the reviewer suggested that there are many other possibilities to explain the results. Therefore, we decided to exclude the experiments and results regarding the mechanistic aspect of Fbl, which was

described originally in Figure 7 in the original version of the manuscript as the reviewer strongly recommended. We are grateful to the reviewer's evaluation of our study as "reviewer feels that ending with a description of the properties of the translationally impacted genes is appropriate and welcome"

Figure R1-1 We transfected E11 NSCs with a bicistronic reporter (BFP-Ezh2-5'UTR-GFP) together with control or BFP siRNA (two different types). After 2 days, we retrieved cells and performed qPCR to investigate the expression of BFP and GFP mRNA. Two different BFP siRNAs present different efficiencies in reduce BFP mRNA and they also reduced GFP mRNA to the same level as BFP mRNA (compare bars with same colors in left and right figures).

Minor concerns:

What do the flow plots for the "positive control" IRES (Cdkn1b) look like? If the authors are claiming that 1-3% GFP signal is sufficient to presume IRES usage, then it is important to show what the %GFP positive cells in their positive control is, and if possible, how it compares to a bona fide viral IRES to get an idea of dynamic range.

Response to Reviewer. We tested a bona fide viral IRES as the reviewer's suggestion. The viral IRES can strongly induce GFP and more than 90% of BFP cells were GFP-positive. We also discussed these concerns with other researchers that study Fbl function in human cells. In their experience, the efficiency of IRES-dependent translation of mRNAs is also lower than 10% (Frédéric CATEZ; personal communication). However, the lower efficiency of the reporter activity compared with viral IRES was not sufficient to explain our *in vivo* experiments. Therefore, we considered that the cap-independent mechanism should be further analyzed in a future study.

Figure R1-2 We transfected E11 NSCs with bicistronic reporter (BFP-virus IRES-GFP) and performed cell sorting after two days. In the transfected cells (BFP+ left panel), 94.5% of these cells also expressed GFP.

Reviewer #3 (Remarks to the Author):

The authors have characterized the temporal gene regulation during neural stem cells progression. The authors focused in the H3K27me3 modifiers, and in particular Fbl. Fbl looks to regulated mRNA translation, likely by recognizing particular sequences in the 5'UTR. Using several state-of-the-art techniques, (e.g. single cell sequencing, ribo-seq, Chip-seq), this work emerges as an interesting example of gene regulation at multiple level, translation, transcription and mRNA level.

I would like to mention some comments that were unclear and/or might improve the work. We would thank the reviewer for positive evaluation of our work and for the critical suggestions.

-Figure 1. Page 5, Lines 110 to 120. It is unclear to me, what is the relation between the changes in H3K27me3 and H3K4me3 and RNA level between E11 to E14. Specially thinking in Fig 1d and how the author selected the 1505 and 20 sites.

Response to Reviewer. The selection of 1505 and 20 sites was based on the statistical analysis. First, we hypothesis that all peaks and their intensity followed a negative binormal distribution among all samples. We obtained parameters using all peaks. Those peaks that did not follow this distribution were considered as peaks that showed differences across samples. By this method, we identified 1505 and 20 sites.

We also verified how expression changed for these genes near these sites. We added these data to the revised Fig. 6k and 6l. We extracted H3K27me3-peaks which were specifically detected in E11 and E14 NPCs, respectively. Then, we annotated these peaks to specific genes using the online tool: GREAT (<http://great.stanford.edu/public/html/>). As a result, we identified 540 and 54 genes whose H3K27me3 peaks were specifically observed in E11 and

E14 NPCs. We have added the genes names in the Supplementary Table 5a,b. As the reviewer suggested in point 6, we have also added GO analysis for H3K27me3. For H3K4me3, as only a few changes could be detected (20 peaks changed), we listed the gene names in Supplementary Table 5e, but did not present their associated GO-terms with H3K4me3.

Figure R1-1 Go term analysis of genes with genomic regions have E11- (a) and E14- (b) specific peaks for H3K27me3.

-Page 7, line 153. The western blot suggesting that Fbl protein is higher in E11 than in E14, is the first experimental data suggesting that translation regulation might be playing a role. Thus, I would show the Western blot as main figure. While I was able to find the blot and the quantification (Extended Data Fig 2c), and one of the replicates does not look great. I was not able to find (Extended Data Fig 2d).

Response to Reviewer. We repeated the experiments and renamed the figure as main Figure 1g and 1h. We confirmed our submitted figures and the Extended Data Fig 2d. There may have been a system error during download. Because we changed the order of the figures as suggested by another reviewer, the previous Extended Data Fig 2 is now Supplementary Figure 2 in the revised version. If the reviewer still cannot find these figures, please contact the editors.

-Across the text, there is almost no statistical information. Just to mention one example, in page 7, line 158 it says “dramatic brain size reduction”, I am not a mouse expert and do not see a “dramatic brain size reduction”. Please quantify it and add a P value. At least in the files that I downloaded; I found the Extended Data Fig 2f but I could not find Extended Data Fig 2g.

Response to Reviewer. We apologize for lack of statistical data. We have added the statistical analysis results in Figure 1j. We verified our submitted figures and find Extended Data Fig 2g was included. There may have been a system error during download.

-Page 11, line 255. The authors found a decreased level of synthesized protein in the DKO NSC (Extended Data fig 6a). Honestly, it is very hard to find the data, there are Extended Data Table 6a and Extended Fig 6a. I am not sure which one I need to see, but in any case, I would expect to find a plot (barplot?) with a P value calculated and not tables. Moreover, I think this is also a very important result and therefore, I would show it as main figure.

Response to Reviewer. We apologize that it was difficult for the Reviewer to find the data. As the reviewer suggested, we have placed the data in the main figure as shown in Figure 4 in the revised manuscript.

Reviewer #4 (Remarks to the Author):

The authors show that H3K4me3 and H3K27me3 modification changes during temporal patterning of NSCs. They show that depletion of Fbl (fibrillarin), a rRNA 2'-O-methyltransferase, results in impaired translation of both the Ezh2 methyltransferase and Kdm6b demethylase of H3K27me3. This engenders a delay in progression of the NSC state, thereby impeding brain development. The authors argue that Fbl selectively enhances the translation of H3K27me3 modifiers via a cap-independent mechanism. Whilst the authors present a substantial body of work, there is much which is missing regarding the mechanism of Fbl-mediated translational control. The discussion is highly speculative, as there is no mention of supportive data in the literature. There are also several issues regarding the text and other concerns outlined below.

Response to Reviewer. Thanks you very much for your precious comments and helpful suggestions. Reviewer 2 also raised concerns about the experiments of Figure 7 and strongly suggested that we remove the reporter assay from current manuscript. We also performed the experiments suggested by both reviewers. Following an extensive discussion with our colleagues about whether we should include these experiment in the revised manuscript, we eventually reach the consensus that this reporter assay was not sufficient to explain the *in vivo* context, and as the reviewer suggested, there are many other possible reasons to explain the results.

Major concerns:

1. The authors claim that Fbl regulates translation of genes involved in H3K27me3 modification in a cap-independent manner (in the heading of the section it is stated “cap-dependent”). The evidence provided by the authors for a cap-independent mechanism is not convincing. First, the data in Fig. 7D should not be presented as a ratio of GFP to BFP signal as one could readily conclude that a lower value in the Fbl siRNA KO condition is simply due to more BFP signal (i.e. cap-dependent translation). This indeed looks like the case for Ezh2 (Extended Fig. 7D). Both GFP and BFP values should be graphed together. Second, there is no dedicated section in the methods detailing how the constructs for each gene were made and how these experiments were performed making it difficult for the reader to interpret the data. Lastly, to convincingly demonstrate that Fbl deletion reduces the translation of Ezh2 and Kdm6b, the authors should perform polysome profiling to show a shift in these mRNAs towards lighter polysome fractions.

They characterize Ezh2 and Kdm6b IRESs only by using bicistronic plasmids, which is prone to misinterpretation [Jackson, R.J. (2013). The current status of vertebrate cellular mRNA IRESs. Cold Spring Harb. Perspect. Biol. 5, pii: a011569]. One common artefact with this approach is the generation of monocistronic mRNAs due to the presence cryptic promoters. Their RT-PCR analysis using primers in BFP and GFP regions is misleading, as it cannot detect such mRNAs (Extended Data, Fig. 7c). Northern blot with a GFP probe or RT-PCR using primers within GFP coding region should be provided. The most rigorous approach would be targeting the BFP cistron coding sequence with siRNA. If this siRNA does not reduce the expression of GFP cistron, then this expression is most likely driven by monocistronic mRNA (Van Eden et al. 2004.RNA 10:720-730). Alternatively, they can generate bicistronic mRNAs and monitor the translation of both cistrons after transfection of these mRNAs into cells. Also, the ability poly(U) to confer upon 5'UTRs an IRES activity is not well studied.

Response to Reviewer. As for first question, to confirm the Fbl influences translation of Ezh2 and Kdm6b, we performed polysome profiling as suggested by the Reviewer. Polysome profiling requires large numbers of cells, but it is difficult to collect large numbers of cells from embryonic brains. Therefore, after fractionating the samples, we pooled the fractions corresponding to the subpolysomes and polysomes and measured the expression levels of mRNA in these pooled fractions. The results showed that when Fbl was knocked down, Ezh2 mRNA was transferred from the polysome to the subpolysome, indicating that Ezh2 translation was affected by the reduction of Fbl (Supplementary Figure 7j). However, because the expression of Kdm6b was relatively low in NSCs, we could not detect any mRNA.

Regarding the second comment, we also used BFP siRNA to test whether GFP expression would be reduced after knockdown of BFP. We found that BFP siRNAs led to similar effect on GFP mRNA (Figure R1-1).

The enrichment of the poly (U) motif the downregulated gene by Fbl-deletion is an observation and we did not explore what this means in this study. To emphasize this, we replaced the sentence from “Moreover, a poly (U) motif was highly enriched in the 5’UTRs of these mRNAs” to “We also noted that 5’UTRs of some of these mRNAs tend to have a unique feature, namely, a poly (U) motif is highly enriched, while the significance of these motif is unknown.”

Figure R1-1 We transfected E11 NSCs with bicistronic reporter (BFP-Ezh2-5’UTR-GFP) together with control or BFP siRNA (two different types). After 2 days, we retrieved cells and performed qPCR to investigate the expression of BFP and GFP mRNA. The two different BFP siRNAs showed different efficiency to reduce BFP mRNA and also reduced GFP mRNA to same level as BFP mRNA (compare bars with same colors in left and right figures).

2. The discussion of the mechanistic aspect of the paper vis-à-vis translation is severely following speculative. For example, “In addition, it is likely that Fbl affects translation via

the structure of 5'UTR of target mRNAs, which restricts the range of translational regulation, eventually generating the specificity of Fbl targets.” It is unclear how an rRNA processing enzyme could affect the structure of the 5'UTR of target mRNAs to promote translation. The authors also have not shown how methylation of rRNA by Fbl confers specificity for the translational regulation of specific mRNAs, as depicted in Fig. 8. The notion that a marginal increase of methylation on some rRNA sites by Fbl (Extended Data Fig. 3d) changes the translational specificity of ribosomes in favour of Ezh2 and Kdm6b IRESs is extremely tenuous, as no experimental support and even mechanistic explanation is provided. Moreover, it is very baffling that the translation from Cdkn 1b, Ezh2 and Kdm6b IRESs is reduced after Fbl knockdown (Figure 7d). In fact, Fbl depletion is supposed to provide a competitive advantage over cellular IRESs, as it inhibits global translation, which is mainly cap-dependent (Extended Data Fig. 6a).

Response to Reviewer. We thank the reviewer’s critical suggestions. We agree that the reviewer that “*It is unclear how an rRNA processing enzyme could affect the structure of the 5'UTR of target mRNAs to promote translation*”. To demonstrate this, the best way was to inhibit the methyltransferase using specific inhibitors. Unfortunately, there was no specific inhibitor for methylation of rRNA. Thus, we could not support our conclusion in this context. Instead, we discussed the potential mechanisms in the Discussion section and modified Figure 8, accordingly.

Your idea that “*Fbl depletion is supposed to provide a competitive advantage over cellular IRESs*” might be based on what is observed during apoptosis where global translation is interrupted and IRES-dependent machinery initiates to translation of apoptosis-related genes. However, there are studies which indicate that knockdown of Fbl in human cells would decrease IRES-independent transcription (Erales et al., 2017). Therefore, the relationship between the global translation and competitive advantage over IRES might be context-dependent.

In addition, reporter assays showed that only 3% of cells could express GFP protein. It is difficult to explain the translation enhanced by Fbl through a cap-independent way mechanism. One reviewer suggested we remove these data, because “the paper would be stronger without trying to prove this specific mechanism – which seems beyond the scope of this paper”. We agree with the reviewer and have removed these data from the Results section. Instead, we added a discussion about the possible mechanisms through which Fbl regulates translation using these data.

Other concerns:

1. Quantification of the brain images (Fig. 2e) should be included to confirm microcephaly.

Response to Reviewer. We had added quantitative data of brain images in revised Fig. 1j.

2. The choice for using EMX1-Cre heterozygous instead of homozygous mice is not clear.

Response to Reviewer. We apologize for the unclear statement. In Emx1-Cre mice, Emx1 is replaced by Cre and the homozygous mouse is also an Emx1-null mouse, in which the corpus callosum is disrupted (Qiu et al., 1996). We have added the description to the manuscript.

3. Fig. 3D, clarification is needed as to what the arrows are representing.

Response to Reviewer. The arrows indicate the temporal and differentiation axes of neural stem cells. We have added the description in figure legend.

4. Fig. 3 E and F, components 1 and 2 need to be described.

Response to Reviewer. We have described these components in the legend of Fig 2 (previous Fig 3 has become Fig 2). The high dimensional transcriptome data of each cell was reduced into 2 components by tSNE.

5. Fig. 4E, the values indicated as percent in the graph are illegible.

Response to Reviewer. We have revised this point accordingly in the revised Fig 3e.

6. The western blot in Fig. 5D is of very poor quality. The columns should be in line and each composite membrane zoomed to the same magnification. These issues apply to other westerns in the manuscript.

Response to Reviewer. We apologize for poor quality of the blots. We have revised the figures accordingly.

7. The authors should include representative images for the O-propargyl-puromycin experiment.

Response to Reviewer. We have added the data to Fig 4.

8. The exons flanked by loxP sites need to be mentioned.

Response to Reviewer. We have added the data. The whole Fbl gene is flanked by LoxP site. We have added a description to Supplementary Figure 2.

9. The dilution for each antibody used should be included in the methods as well as the final concentration of protease inhibitor used for western blot.

Response to Reviewer. We have added the information in the Methods section.

10. The authors should provide a brief description as to how the 10X-Genomics single cell seq works and provide a reference if it has been successfully used in previous studies.

Response to Reviewer. We have added description for the 10X Genomics single cell seq. We also compared our data with previous data to show the high quality of our data in Supplementary Figure 1.

11. The manuscript requires thorough editing as there are many typos and grammatical errors.

Response to Reviewer. We apologize for the numerous typos and errors. We will send the revised manuscript to an English editing service for correction.

s

12. Line spacing should be the same throughout the main text.

Response to Reviewer. We have revised this point.

Reference:

Erales, J. *et al.* Evidence for rRNA 2' -O-methylation plasticity: Control of intrinsic translational capabilities of human ribosomes. *Proc Natl Acad Sci USA* **114**, 12934–12939 (2017).

Qiu, M. *et al.* Mutation of the Emx-1 homeobox gene disrupts the corpus callosum. *Dev. Biol.* **178**, 174–178 (1996).

REVIEWERS' COMMENTS

Reviewer #1 (Remarks to the Author):

This revised manuscript, accompanied data, and responses to reviewers have much strengthened the manuscript. This reviewer appreciates the authors' effort and has 2 points:

- R1-pt 4, text line 349-354: authors may consider changing "we observed genes that regulated of proliferation of NSCs..." to "we observed genes that regulate the proliferation of NSCs..."
- R1-pt11, the authors made a good effort in responding to the temporal and differentiation defects. Because the title of the manuscript clearly emphasizes the potential temporal programming role of Fbl, the authors appear to prefer the temporal programming regulation over the differentiation program regulation by Fbl. The authors would need to make a stronger attempt to support this preference. This issue also relates to R1-pt 13. To suggest that the differentiation defects relate to delay temporal progression, the DKO cells would need to eventually 'reach' the appropriate differentiation stage. The authors would need to provide this piece of data or de-emphasize the temporal programming role of Fbl.

Reviewer #2 (Remarks to the Author):

The authors did a great job of responding to all reviewer comments. Thanks for your efforts. At this time, I believe the paper is suitable for publication.

Reviewer #4 (Remarks to the Author):

the authors have performed many new experiments. They responded satisfactorily to most of my concerns.

There are a few issues remaining:

1. Does the GO term "cytosolic part" mean all genes in the cytosol? Please clarify.
2. The quantification in Fig. 1J should be mm² and not cm² on the y-axis.
3. There are still many typos, unfortunately.

REVIEWERS' COMMENTS

Reviewer #1 (Remarks to the Author):

This revised manuscript, accompanied data, and responses to reviewers have much strengthened the manuscript. This reviewer appreciates the authors' effort and has 2 points:

We appreciate the reviewer's effort to read the manuscript and thank the reviewer for the constructive suggestions.

- R1-pt 4, text line 349-354: authors may consider changing “we observed genes that regulated of proliferation of NSCs...” to “we observed genes that regulate the proliferation of NSCs...”

We changed that point.

- R1-pt11, the authors made a good effort in responding to the temporal and differentiation defects. Because the title of the manuscript clearly emphasizes the potential temporal programming role of Fbl, the authors appear to prefer the temporal programming regulation over the differentiation program regulation by Fbl. The authors would need to make a stronger attempt to support this preference. This issue also relates to R1-pt 13. To suggest that the differentiation defects relate to delay temporal progression, the DKO cells would need to eventually ‘reach’ the appropriate differentiation stage. The authors would need to provide this piece of data or de-emphasize the temporal programming role of Fbl.

We agreed with the reviewer's suggestion. Because Fbl has dual functions in both regulation of temporal programming and differentiation. We weakened the expression of the temporal programming role of Fbl and equally emphasized the role of Fbl in differentiation programming by following changes:

First, we changed our title as “Selective translation of epigenetic modifiers affects the temporal pattern and differentiation of neural stem cells”. Second, we emphasized the function of Fbl in differentiation of neural stem cells (NSCs) in the manuscript. As the editor suggested us to summarize our results at end of introduction part, we added the “Here, we show that Fbl, an rRNA methyltransferase, is required for temporal progression and differentiation of NSCs.” (line 85-86). In the result section, we also added the description “These results suggest that “Fbl has dual functions required for temporal progression and differentiation of NSCs.” (line 229)

Reviewer #2 (Remarks to the Author):

The authors did a great job of responding to all reviewer comments. Thanks for your

efforts. At this time, I believe the paper is suitable for publication.

We thank the reviewer for careful reading of our manuscript and for their constructive suggestions.

Reviewer #4 (Remarks to the Author):

the authors have performed many new experiments. They responded satisfactorily to most of my concerns.

We thank the reviewer for careful reading of our manuscript and for their constructive suggestions.

There are a few issues remaining:

1. Does the GO term "cytosolic part" mean all genes in the cytosol? Please clarify. GO term cytosolic part means "The part of the cytoplasm that does not contain organelles but which does contain other particulate matter, such as protein complexes". To explicitly show the meaning of this GO term, we listed the genes containing in the GO term of "cytosolic part" in Supplementary Data 1c.

2. The quantification in Fig. 1J should be mm² and not cm² on the y-axis. We apologize for our mistake. We corrected it.

3. There are still many typos, unfortunately.

We apologize for typos, although we submitted to professional English editing (which is recommended by the Nature group) before the previous submission of the revised manuscript. We corrected these typos in the current figures and manuscript as much as possible.